# Genome analysis of biosurfactant producing bacterium, *Bacillus tequilensis*

**Anuraj Nayarisseri**[1,2,3]*, **Sanjeev Kumar Singh**[1,4]*

**1** Computer Aided Drug Designing and Molecular Modeling Lab, Department of Bioinformatics, Alagappa University, Karaikudi, Tamil Nadu, India, **2** In silico Research Laboratory, Eminent Biosciences, Indore, Madhya Pradesh, India, **3** Bioinformatics Research Laboratory, LeGene Biosciences Pvt Ltd., Indore, Madhya Pradesh, India, **4** Department of Data Sciences, Centre of Biomedical Research, Lucknow, India

\* anuraj@eminentbio.com (AN); skysanjeev@gmail.com (SKS)

**Data Availability Statement:** All relevant data are within the manuscript and its Supporting Information files.

**Funding:** The author(s) received no specific funding for this work.

## Abstract

Bioremediation is crucial for recuperating polluted water and soil. By expanding the surface area of substrates, biosurfactants play a vital role in bioremediation. Biosurfactant-producing microbes release certain biosurfactant compounds, which are promoted for oil spill remediation. In the present investigation, a biosurfactant-producing bacterium *Bacillus tequilensis* was isolated from Chilika Lake, Odisha, India (latitude and longitude: 19.8450 N 85.4788 E). Whole-Genome Sequencing (WGS) of *Bacillus tequilensis* was carried out using Illumina NextSeq 500. The size of the whole genome of *Bacillus tequilensis* was 4.47 MB consisting of 4,478,749 base pairs forming a circular chromosome with 528 scaffolds, 4492 protein-encoding genes (ORFs), 81 tRNA genes, and 114 ribosomal RNA transcription units. The total raw reads were 4209415, and the processed reads were 4058238 with 4492 genes. The whole genome obtained from the present investigation was used for genome annotation, variant calling, variant annotation, and comparative genome analysis with other existing *Bacillus* species. In this study, a pathway was constructed which describes the biosurfactant metabolism of *Bacillus tequilensis*. The study identified that genes such as *SrfAD*, *SrfAC*, *SrfAA* and *SrfAB* are involved in biosurfactant synthesis. The sequence of the genes *SrfAD*, *SrfAC*, *SrfAA*, *SrfAB* was deposited in GenBank database with accession MUG02427.1, MUG02428.1, MUG02429.1, MUG03515.1 respectively. The whole genome sequence was submitted to GenBank with an accession RMVO00000000 and the raw fastq reads were submitted to SRA, NCBI repository with an accession: SRX5023292.

## 1. Introduction

Heavy metal contamination has now become a serious ecological threat raising environmental concerns. Metals especially cadmium and zinc have posed a serious threat as their degradation to innocuous products is hard and takes millions of years [1–3]. Bioremediation systems which have been long proposed to neutralize metal contamination, however, have low bioavailability leading to an incomplete bioremediation process. Further, such bioremediation processes like phytoremediation with synthetic chelators are proven to be expensive and

**Competing interests:** The authors have declared that no competing interests exist.

environmentally hazardous [4, 5]. Various surface-active compounds (SACs) commonly bio-surfactants produced by microorganisms have emerged as safe alternatives to chemical reme-diation [6–8].

The Whole-genome sequence represents a valuable shortcut, helping scientists to find genes much more easily and quickly. It is expected that being able to study the entire genome sequence will help in understanding how the genes endeavor together to direct the mainte-nance, development, and growth of a whole organism. Besides, it can use to predict the genes involved in the synthesizing of biosurfactants in microbes [9, 10]. Therefore, the present study aimed to sequence the whole genome of biosurfactant-producing *Bacillus tequilensis* using Next-Generation sequencing, De-novo assembly, genome annotation, variant calling, and var-iant annotation.

## 2. Results and discussion

### 2.1 Identification of biosurfactant producing *Bacillus tequilensis*

The majority of biosurfactants are produced by the microbes such as the *Pseudomonas* genus followed by *Bacillus* and *Acinetobacter* respectively [11]. In a previous investigation, a novel strain of *Bacillus tequilensis* was identified by various biochemical tests, microbial tests such as the Haemolysis test, oil spreading test, CTAB agar plate test, Drop collapse test, etc and the identification of the novel bacterial strain was performed by molecular characterization i.e 16S rRNA gene sequencing and phylogenetic assessment. The sequence of the bacteria, which was found to be novel, it was given the name *Bacillus tequilensis* strain ANSKLAB04 and deposited in GenBank with the accession number KU529483 [12]. The same novel strain was considered and employed for the current investigation of whole genome sequencing. In the present inves-tigation, we have also concluded the novel strain by Average Nucleotide Identity (ANI) analy-sis and whole genome to genome comparison studies. According to Average Nucleotide Identity (ANI) analysis and digital DNA-DNA hybridization, *Bacillus tequilensis's* genome sequence was found to be more similar to *Bacillus subtilis* by 98.56% of Ortho ANI, 98.47% Original ANI, and Genome to Genome Distance count (GGDC) of 0.0146. Lower GGDC indi-cates a closer relationship and less gap (distance) between the species. *Bacillus halotolerans* and *Bacillus tequilensis* ANSKSLAB04 were found to have the second-highest similarity, with an Orotho ANI of 96.02%, an Original ANI of 95.98%, and a GGDC distance of 0.04 respectively. Similarly, next three similar species are *Bacillus tequilensis* KCTC 13622, *Bacillus vallismortis* and *Bacillus mojavensis* RO-H-1 = KCTC 3706 which are closely related to *Bacillus tequilensis* ANSKLAB04 [Fig 1] [Table 1].

### 2.2 Bioanalyzer profile

The DNA isolation was performed using Phenol/Chloroform (PCl) genomic DNA extraction method [12]. The bioanalyzer profile of the prepared WGS library showed fragments in a size range of 300-600bp. The effective insert size of the library was 180-480bp flanked by adaptors having a combined size of ~120bp. Based on the fragment distribution and concentration, the library was suitable for sequencing using Illumina platform.

### 2.3 Genome representation

The complete genome of *Bacillus tequilensis* consists of a single circular chromosome of 4,478,749 bp with an average G+C content of 46.33% (**Table 2** and **S1 Table**). The 4492 pre-dicted coding ORFs cover 87% of the complete genome, and each ORF has a moderate length of 283 aa (**S1 Table**). Among these, 1,347, i.e. 67.4% were assigned as putative functions, 258,

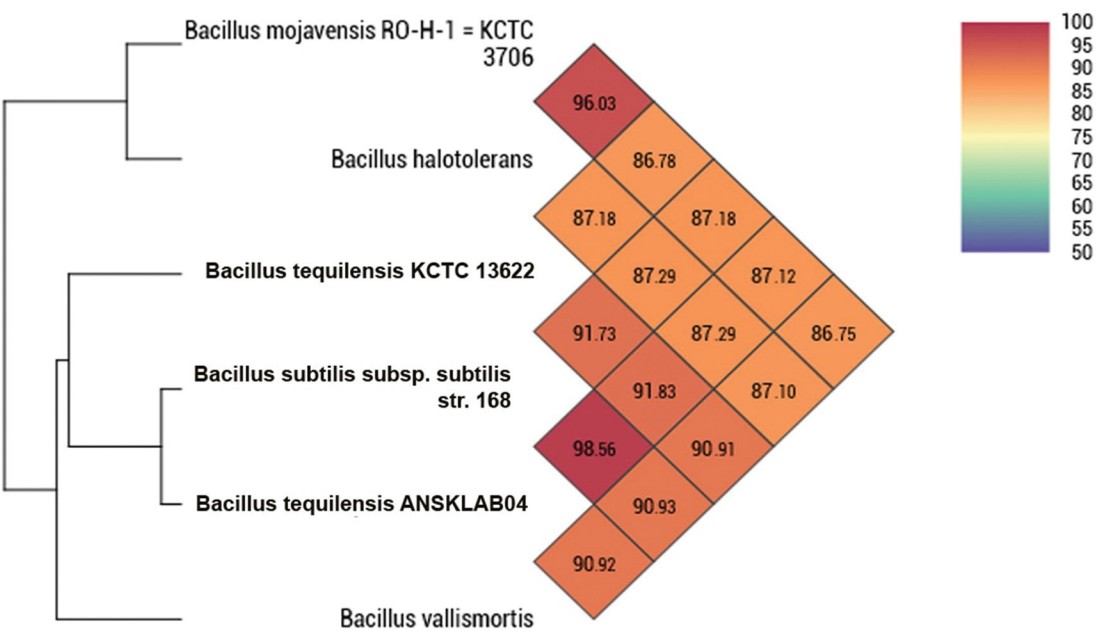

**Fig 1. Average Nucleotide Identity (ANI) of *Bacillus tequilensis* compared to the top 5 homologous species.**

i.e. 12.9% matched to sustain hypothetical coding sequences of an anonymous function, and the rest 394, i.e. 19.7% shows no similarities to any known genes [Table 3].

All genes were classified according to the COG classification. http://www.ncbi.nlm.nih.gov/COG/

The variations in the nucleotide frequencies across the whole genome sequence were investigated using a non-overlapping active platform and by framing three indices of nucleotide frequency: G+C%, (G+C)/(A+T+C+G), divergence from [A] = [T], (A-T)/(A+T), and divergence from [C] = (G), (C-G)/(C+G). These 3 indices are, by representing, pairwise-independent and summarize relative nucleotide frequencies without loss of information. Because of their very

**Table 1. Average Nucleotide Identity (ANI) of *Bacillus tequilensis* compared to the top 5 homologous species.**

| Genome 1 | Genome 2 | OrthoANI value (%) | -Original ANI value (%) | GGDC distance |
|---|---|---|---|---|
| *Bacillus mojavensis* RO-H-1 = KCTC 3706 | *Bacillus tequilensis* KCTC 13622 | 86.7789 | 86.6014 | 0.131902482 |
| *Bacillus mojavensis* RO-H-1 = KCTC 3706 | *Bacillus halotolerans* | 96.0287 | 95.986 | 0.04069134 |
| *Bacillus mojavensis* RO-H-1 = KCTC 3706 | *Bacillus subtilis* subsp. subtilis str. 168 | 87.1754 | 87.026 | 0.128355066 |
| *Bacillus mojavensis* RO-H-1 = KCTC 3706 | *Bacillus tequilensis* ANSKLAB04 | 87.1174 | 86.9002 | 0.129689628 |
| *Bacillus mojavensis* RO-H-1 = KCTC 3706 | *Bacillus vallismortis* | 86.753 | 86.5856 | 0.132413757 |
| *Bacillus tequilensis* KCTC 13622 | *Bacillus halotolerans* | 87.1795 | 86.9702 | 0.128263844 |
| *Bacillus tequilensis* KCTC 13622 | *Bacillus subtilis* subsp. subtilis str. 168 | 91.7345 | 91.4396 | 0.084283969 |
| *Bacillus tequilensis* KCTC 13622 | *Bacillus tequilensis* ANSKLAB04 | 91.8291 | 91.6312 | 0.083354855 |
| *Bacillus tequilensis* KCTC 13622 | *Bacillus vallismortis* | 90.9057 | 90.6751 | 0.092500797 |
| *Bacillus halotolerans* | *Bacillus subtilis* subsp. subtilis str. 168 | 87.2852 | 87.1169 | 0.126449587 |
| *Bacillus halotolerans* | *Bacillus tequilensis* ANSKLAB04 | 87.2934 | 87.0496 | 0.128261295 |
| *Bacillus halotolerans* | *Bacillus vallismortis* | 87.0996 | 86.9208 | 0.128734593 |
| *Bacillus subtilis* subsp. subtilis str. 168 | *Bacillus tequilensis* ANSKLAB04 | 98.5622 | 98.4731 | 0.014632364 |
| *Bacillus subtilis* subsp. subtilis str. 168 | *Bacillus vallismortis* | 90.9318 | 90.9207 | 0.089900259 |
| *Bacillus tequilensis* ANSKLAB04 | *Bacillus vallismortis* | 90.9179 | 90.8406 | 0.090797464 |

**Table 2. Assembly statistics of scaffolds.**

| Assembly Stat | Assembly |
|---|---|
| Contigs Generated | 528 |
| Maximum Contig Length | 1664507 |
| Minimum Contig Length | 500 |
| Average Contig Length | 8482 |
| Median Contig Length | 597 |
| Total Contigs Length | 4478749 |
| Total Number of Non-ATGC Characters | 510 |
| Percentage of Non-ATGC Characters | 0.011 |
| Contigs > = 100 bp | 528 |
| Contigs > = 200 bp | 528 |
| Contigs > = 500 bp | 528 |
| Contigs > = 1 Kbp | 66 |
| Contigs > = 10 Kbp | 12 |
| Contigs > = 1 Mbp | 2 |
| N50 value | 1077242 |

low frequency, ambiguous nucleotide bases were not taken into account. The SD (standard deviation) for the 3 indices is given by

$$SD[(G + C)/(A + T + C + G)] = \frac{1}{N}\sqrt{\frac{SW}{N}} \tag{1}$$

$$SD[(A - T)/(A + T)] = \frac{2}{W}\sqrt{\frac{AT}{W}} \tag{2}$$

$$SD[(C - G)/(C + G)] = \frac{2}{S}\sqrt{\frac{CG}{S}} \tag{3}$$

Where, $W = A + T$, $S = C + G$ and $N = A + T + C + G$.

Normal distribution approximation was used as the total numbers of bases were large. The strand analyzed here was the 5' to 3' strand clockwise on the genetic map. A window size of 1 kb was used. From the inside: green and red bars represent RNA sequences on positive and negative strands respectively. Circle 1, represents G + C content (window size: 10Kb) higher and lower than 45%, where red represents higher and green represents lower. Circles 2:—represents GC skewness, where the green and red represents positive and negative value respectively [Fig 2].

## 2.4 Gene ontology and biological annotation

The gene ontology analysis concluded that 18.99% of genes in *Bacillus tequilensis* belonged to transferase activity, 13.55% of genes belonged to kinase activity, 9.3% of genes were involved in ATP binding, 9.3% genes were involved with hydrolase activity, 6.91% genes were involved in methyl transferase activity, 5.98% of genes were associated with lipase activity, 5.98% of genes were involved in oxidoreductase activity, 4.9% of genes were in lyase activity, 3.05% genes were involved in peptidase activity, whereas only 2.79% genes were involved in cell division, 2.9% genes were in carbohydrate transport, 7.7% genes were in ribose production, and only 3.8% genes were involved in viral capsid [Fig 3].

**Table 3. Functional categories of predicted genes in *Bacillus tequilensis* genome.**

| COG categories | No of Genes |
|---|---|
| **Information storage and processing** | |
| J. Translation, ribosomal structure and biogenesis | 245 |
| A. RNA processing and modification | 25 |
| K. Transcription | 231 |
| L. Replication, recombination and repair | 238 |
| B. Chromatin structure and dynamics | 19 |
| **Cellular Process** | |
| D. Cell cycle control, cell division, chromosome partitioning | 72 |
| Y. Nuclear structure | 2 |
| V. Defense mechanisms | 46 |
| T. Signal transduction mechanisms | 152 |
| M. Cell wall/membrane/envelope biogenesis | 188 |
| N. Cell motility | 96 |
| Z. Cytoskeleton | 12 |
| W. Extracellular structures | 1 |
| U. Intracellular trafficking, secretion, and vesicular transport | 158 |
| O. Posttranslational modification, protein turnover, chaperones | 203 |
| **Metabolism** | |
| C. Energy production and conversion | 258 |
| G. Carbohydrate transport and metabolism | 230 |
| E. Amino acid transport and metabolism | 270 |
| F. Nucleotide transport and metabolism | 95 |
| H. Coenzyme transport and metabolism | 179 |
| I. Lipid transport and metabolism | 94 |
| P. Inorganic ion transport and metabolism | 212 |
| Q. Secondary metabolites biosynthesis, transport and catabolism | 88 |
| **Poorly characterized** | |
| R. General function prediction only | 702 |
| S. Function unknown | 1347 |
| Not in COG | |

## 2.5 Subsystem classification

Genes obtained from the whole genome of *Bacillus tequilensis* have been used for the classification of the subsystem. Subsystems were categorized based on the cofactors, cell wall, virulence metabolism, potassium metabolism, membrane transport, iron acquisition and metabolism, RNA metabolism, cell division, and cell cycle, motility, and chemotaxis, fatty acids, lipids and isoprenoids, nitrogen metabolism, etc were discussed in [**S2 Table**] [Fig 4].

## 2.6 Metabolism of biosurfactant producing genes

*Bacillus tequilensis* produces a biosurfactant that belongs to the class of lipopeptides having excellent emulsifying properties and was capable of reducing the surface tension of water to a significantly lower value. The genes associated with producing biosurfactants are listed in [Table 4]. Among the several different classes of biosurfactant-producing bacteria genera, the members of the genera *Bacillus* or *Pseudomonas*, due to their wide range of applications and resourcefulness can be more often used. *Bacillus* species are phenotypically and genotypically heterogeneous. Based on several investigations, a unique inhabitant of *Bacillus* sp. found at the

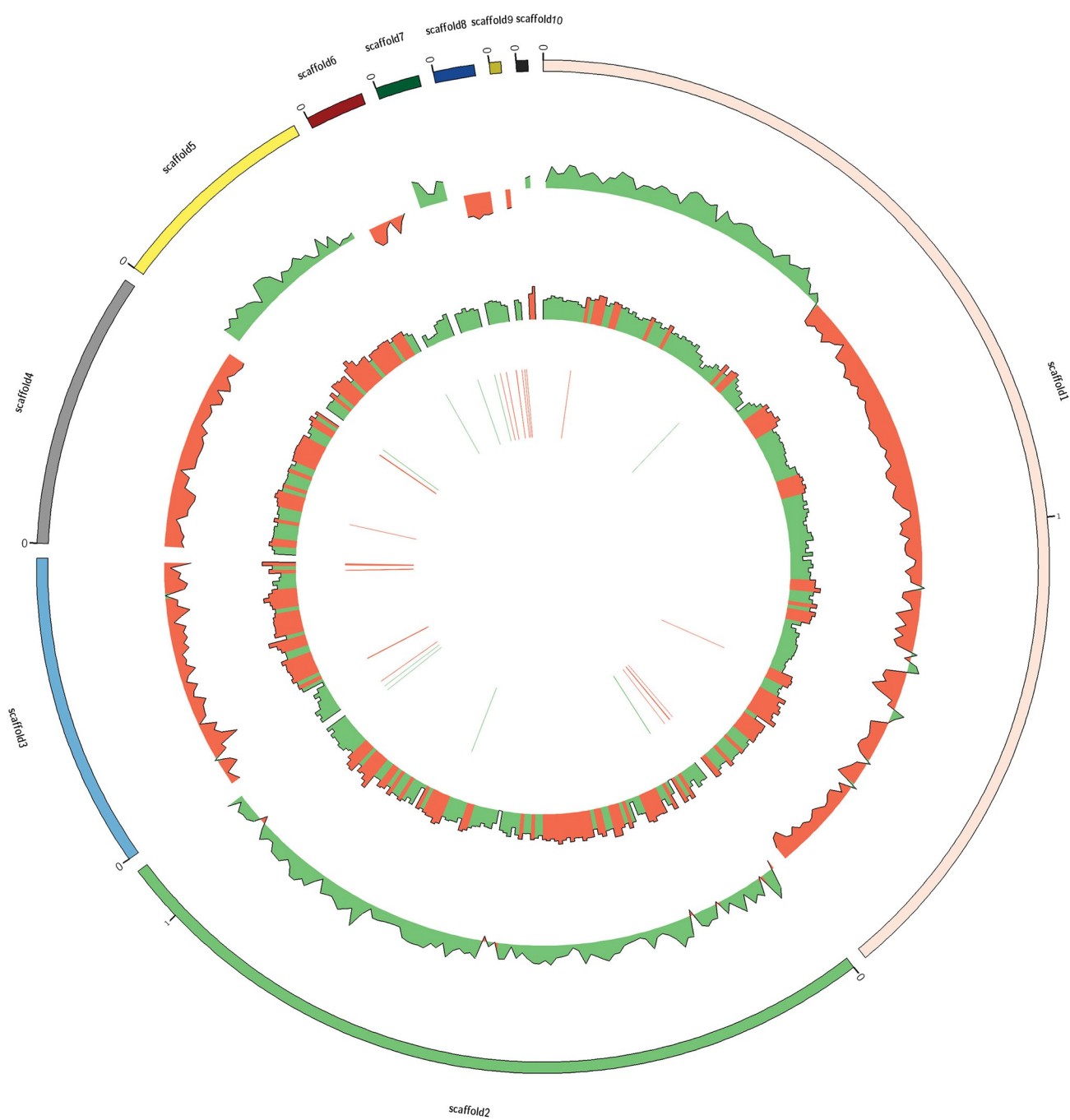

**Fig 2. Genome map of *Bacillus tequilensis*.**

marine site such as *B. subtilis*, *B. licheniformis*, *B. cereus*, *B. amyloliquefaciens*, *B. pumilus, and B. mycoides*. *Bacillus subtilis* produces a lipopeptide biosurfactant called surfactin, which is coded by four ORFs named *SrfA*, *SrfB* (also known as *ComA*), *SrfC*, and *SrfD*. The *sfp* gene is considered an essential component of peptide synthesis systems and plays a major role in the regulation of surfactin biosynthesis and gene expression. *Srf* gene amplification is at 268 bp whereas the expression of the *sfp* gene is amplified at 675 bp [13]. The peptide synthesizes for

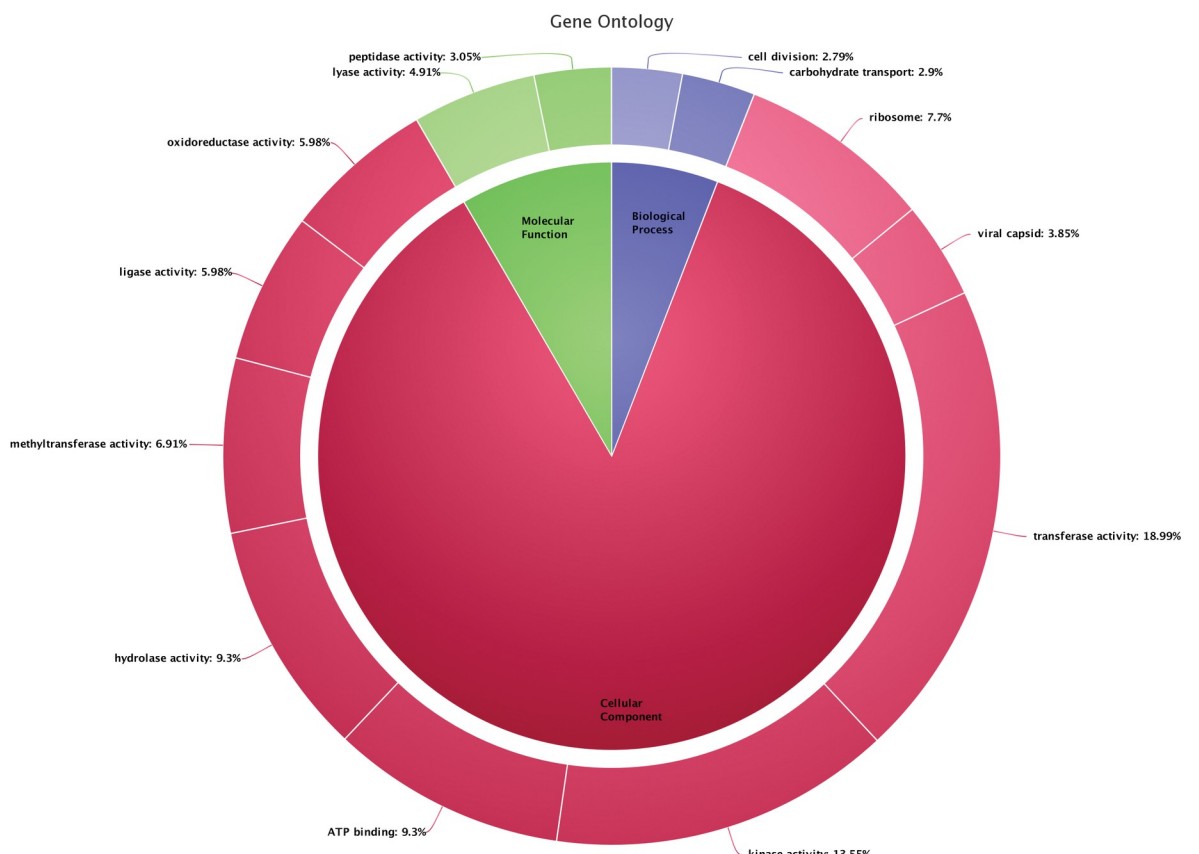

**Fig 3. Biological annotation of *Bacillus tequilensis* ANSKLAB04.**

an amino acid moiety of surfactin is encoded by four ORFs in the *srfA* operon namely *SrfAA*, *SrfAB*, *SrfAC*, and *SrfAD*, *SrfA-TE*, and also contains *comS* gene lying within the out-of-frame with the *srfB* [14]. Porob S. et al. 2013 and Nakano M. et al., 1992 isolated the *SrfA* gene from *Bacillus* amplified at 580 bp, and the authors concluded the biological significance of the *SrfA* gene in biosurfactant production [15–17]. From *Bacillus tequilensis* we identified the *SrfA* which is involved in biosurfactant production and the sequence of the *SrfA*(242 aa) was deposited in GenBank with accession MUG02427.1.

Besides, lichenysin is another lipopeptide biosurfactant produced by *B. licheniformis* coded by lichenysin operon (*LchA*) and comprises four peptide synthetase genes: *LicAA*, *LicAB*, *LicAC*, and *LicAD*. In another study, the authors isolated genes sfp (Phosphopantetheinyl transferase 224 amino acids) and mapped at 4kb downstream to operon *srfA*, and the authors also concluded it is essential for the post-translational changes to surfactin synthetase in microbes [15, 16]. In this study, we have identified *sfp* gene from *Bacillus tequilensis* and the sequence of the *sfp* (Phosphopantetheinyl transferase 224 amino acids) was deposited in GenBank with accession MUG02422.1.

Moreover, two operons, *srfA* and *pps* were found to be present in UMX-103 and *B. subtilis* 168 strains only involved in biosurfactant synthesis. The *srfA* operon contains four genes such as *srfAA*, *srfAB*, *srfAC*, and *srfAD* and the operon pps contains four genes named as *ppsB*, *ppsC*, *ppsD*, *and ppsE*. The genes, *rmlA*, *rmlB*, *rmlC*, and *rmlD* are only present in UMX -103 strains whereas, *sigA*, *DnaK*, *and LytR* are present specifically in *Bacillus* strain. Besides, the genes *comA*, *comP*, *rpoN*, *abrB*, and *ResD* are presented in both UMX-103 and *B. subtilis* 168

## Subsystem Category Distribution

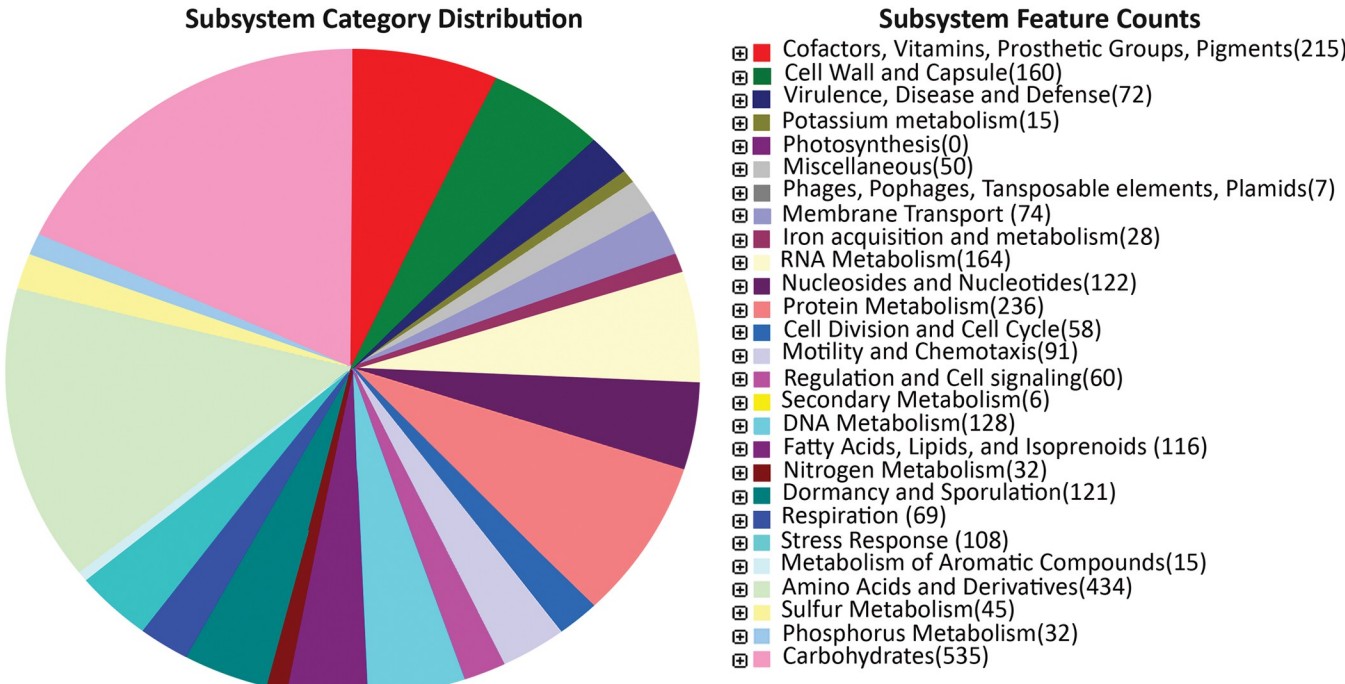

**Subsystem Feature Counts**

- ⊞ ■ Cofactors, Vitamins, Prosthetic Groups, Pigments(215)
- ⊞ ■ Cell Wall and Capsule(160)
- ⊞ ■ Virulence, Disease and Defense(72)
- ⊞ ■ Potassium metabolism(15)
- ⊞ ■ Photosynthesis(0)
- ⊞ ■ Miscellaneous(50)
- ⊞ ■ Phages, Pophages, Tansposable elements, Plamids(7)
- ⊞ ■ Membrane Transport (74)
- ⊞ ■ Iron acquisition and metabolism(28)
- ⊞ ■ RNA Metabolism(164)
- ⊞ ■ Nucleosides and Nucleotides(122)
- ⊞ ■ Protein Metabolism(236)
- ⊞ ■ Cell Division and Cell Cycle(58)
- ⊞ ■ Motility and Chemotaxis(91)
- ⊞ ■ Regulation and Cell signaling(60)
- ⊞ ■ Secondary Metabolism(6)
- ⊞ ■ DNA Metabolism(128)
- ⊞ ■ Fatty Acids, Lipids, and Isoprenoids (116)
- ⊞ ■ Nitrogen Metabolism(32)
- ⊞ ■ Dormancy and Sporulation(121)
- ⊞ ■ Respiration (69)
- ⊞ ■ Stress Response (108)
- ⊞ ■ Metabolism of Aromatic Compounds(15)
- ⊞ ■ Amino Acids and Derivatives(434)
- ⊞ ■ Sulfur Metabolism(45)
- ⊞ ■ Phosphorus Metabolism(32)
- ⊞ ■ Carbohydrates(535)

**Fig 4. Subsystem category distribution of *Bacillus tequilensis* ANSKLAB04.**

[18]. Based on the above literature biological annotation, we have identified *DnaK* and *LytR* genes from *Bacillus tequilensis*, and the sequence was deposited in NCBI with accession MUF99480.1 and MUG01692.1 respectively.

*Pseudomonas* species required Plasmid-encoded- *rhlA*, B, R and I genes of rhl quorum-sensing system for the production of glycolipid biosurfactants as well as also involved in the production of rhamnolipids in a heterologous host. Iturin A is an antifungal lipopeptide bio-surfactant produced by certain *Bacillus subtilis* strains such as *Bacillus subtilis* RB14 is com-posed of four ORF namely *ituD*, *ituA*, *ituB*, and *ituC*, whose disruption leads to specific deficiency in iturin A production. The three genes of arthrofactin operon of *Pseudomonas* namely *arfA*, *arfB*, and *arfC* encode *ArfA*, *ArfB*, *and ArfC* containing two, four, and five func-tional modules respectively required for condensation, adenylation and thiolation. Besides, Amphisin is produced by *Pseudomonas* sp. DSS73 requires *gacS* and *amsY* genes for the pro-duction of biosurfactant as these genes are mutants defective in the genes. Amphisin synthesis is regulated by the *gacS* gene as the *gacS* mutant regains the property of surface motility upon the introduction of a plasmid. Moreover, genes *dnaK*, *dnaJ*, and *grpE* positively regulate the biosynthesis of putisolvin [14]. Putisolvin biosynthesis genes such as *dnaK*, *dnaJ*, and *grpE* from *Bacillus tequilensis* were identified and the sequence was deposited in GenBank with accession MUF99480.1 MUF99481.1, MUF99479.1 respectively.

*Acinetobacter* species produces high molecular weight biosurfactants—Emulsan and Alasan with the involvement of gene. *AlnA*, *AlnB* and *AlnC* are essential for Alasan biosynthesis whereas *wza*, *wzb*, *wzc*, *wzx*, and *wzy* are required for Emulsan biosynthesis. For the produc-tion of fungal biosurfactants, *emt1* and *cyp1* are the two genes involved in the synthesis of these glycolipids, and *fb1* and *hfb2* genes regulate the synthesis of hydrophobin [14]. Thus, gene plays a major role in the biosynthesis of various microbial surfactants, and hence the role of molecular genetics and gene regulation mechanisms in the production of biosurfactant is

Table 4. Biosurfactants producing genes of *Bacillus* species.

| S.No | Gene involved in Biosurfactant production | Reference |
|------|-------------------------------------------|-----------|
| 1. | *srf* | [13] |
| 2. | *sfp* | [13–15] |
| 3. | *srfA* | [16] |
| 4. | *rhl*B | [16] |
| 5. | *cfp* | [17] |
| 6. | *srfAA* | [18] |
| 7. | *srfB* | [18] |
| 8. | *srfAB* | [18] |
| 9. | *srfAD* | [18] |
| 10. | *Spf* | [18] |
| 11. | *ppsB* | [18] |
| 12. | *ppsC* | [18] |
| 13. | *ppsD* | [18] |
| 14. | *X ppsE* | [18] |
| 15. | *X dhbF* | [18] |
| 16. | *X rmlA* | [18] |
| 17. | *X rmlB* | [18] |
| 18. | *X rmlC* | [18] |
| 19. | *X rmlD* | [18] |
| 20. | *X comA* | [18] |
| 21. | *X comP* | [18] |
| 22. | *X ResD* | [18] |
| 23. | *X LiaR* | [18] |
| 24. | *spo0A* | [18] |
| 25. | *rpoN* | [18] |
| 26. | *X crsA* | [18] |
| 27. | *X sigA* | [18] |
| 28. | *abrB* | [18] |
| 29. | *X DnaK* | [18] |
| 30. | *LytR* | [18] |

essential. In this study, we have identified biosurfactant-producing genes and corresponding ORFs of *Bacillus tequilensis* such as gene *SrfAD*, *SrfAC*, *SrfAA*, and the sequence of the same was deposited in GenBank database with accession MUG02427.1, MUG02428.1, MUG02429.1, MUG03515.1 respectively.

## 2.7 Biosurfactant / Lipopeptide metabolism of *Bacillus tequilensis*

Considering the biosurfactant-producing genes described in various literature, we classified the genes of *Bacillus tequilensis* based on the established efficient biosurfactant activity and broad applications. Biosurfactant is proven to be promising; possessing unique properties of low toxicity and higher biodegradability. In the present investigation, we constructed a pathway that describes the biosurfactant metabolism of *Bacillus tequilensis* [Fig 5]. The lipopeptide synthesized constitutes a long chain of fatty acids along with glutamate acid (Glu), leucine (Leu), aspartic acid (Asp), and valine (Val). The synthesis is non-ribosomal by a large multienzyme peptide, non-ribosome peptide synthases (NRPS). The peptide synthetase required for an amino acid moiety of surfactin is encoded by four open reading frames in the *srfA* operon

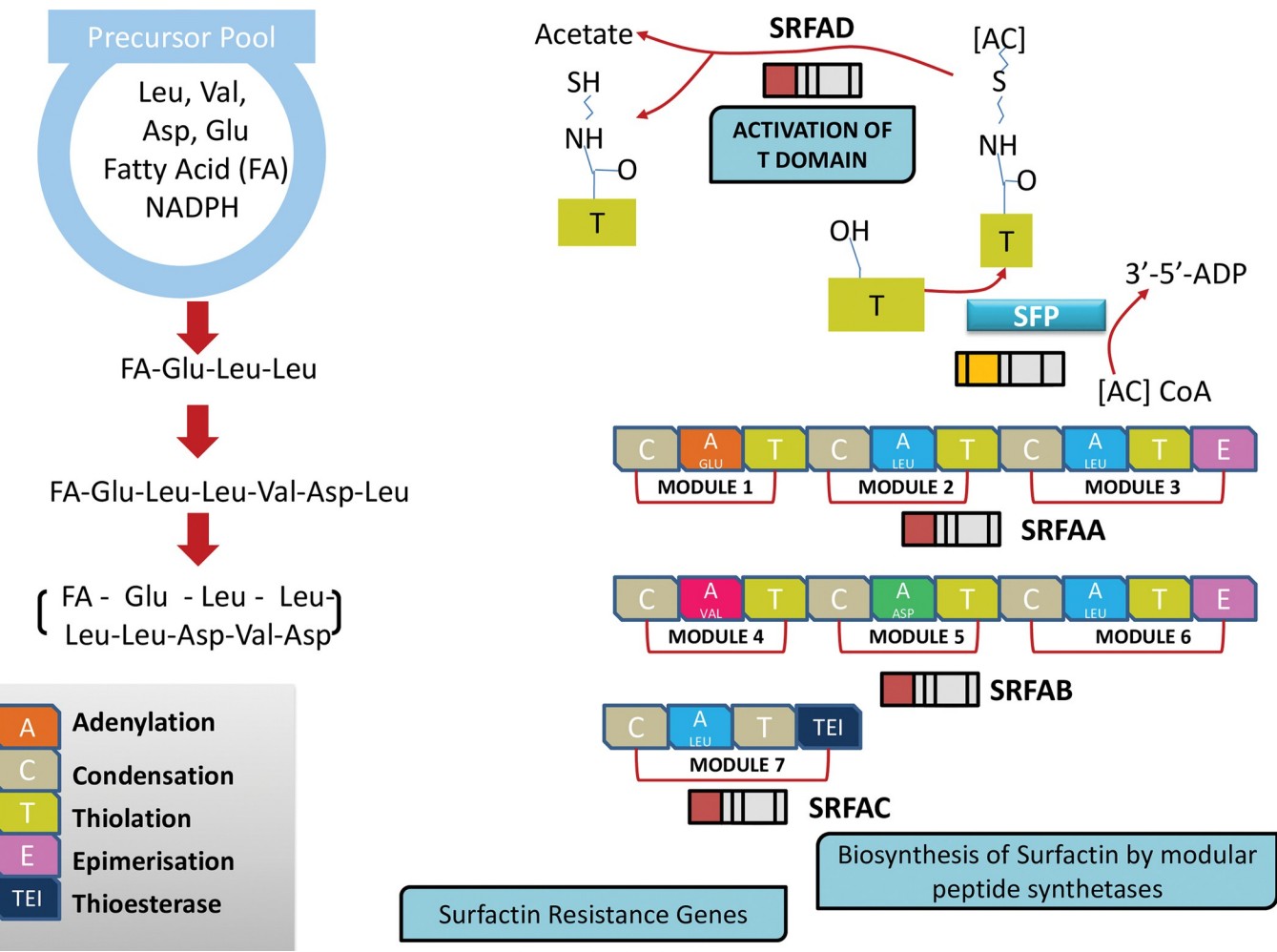

**Fig 5. Biosurfactant / Lipopeptide metabolism of *Bacillus* species.**

namely *SrfAA*, *SrfAB*, *SrfAC*, *and SrfAD or SrfA-TE. SrfA*, *SrfB*, *SrfC*, and s*rfD* constitute the four main enzymes for surfactin formation. *SrfD* is the most important enzyme as it initiates surfactin formation. *srfA* operon plays an important role in post-translational modifications to surfactin synthetase.

Different modules have been marked based on different pathways involved in the synthesis of biosurfactants, such as glycolysis, TCA cycle, NADPh generation, amino acid biosynthesis, fatty acid synthesis, and synthesis of surfactin. Seven modules represent the different pathways required for the production of glutamate acid (Glu), leucine (Leu), aspartic acid (Asp), and valine (Val). The precursors for biosynthesis of Val/Leu, Glu/ Asp, and fatty acids are the product of glycolysis and TCA cycle such as pyruvate, 2-oxo-glutarate, oxaloacetate, and acetyl-CoA. The genes of *Bacillus tequilensis* involved in the utilization of sucrose, including *sacP*, *murP*, and *sacA*, which encode a sugar transporter, permease, and sucrose-6-phosphate hydrolase, were identified and the sequence was deposited in GenBank with accession MUF99868.1, MUG00557.1, MUG01465.1 respectively. The NADPH generation and pentose are produced by the pentose phosphate pathway catalyzed by *zwf* and GNDA enzymes.

The biosynthesis of Glu, Asp, Val, and Leu, are considered as the intrinsic components of surfactin. Glu/Asp are synthesized by aspartate aminotransferases such as AspB and YhdR

were identified from *Bacillus tequilensis* and the sequence was deposited to GenBank using accession MUF99794.1 and MUF99877.1respectively. The efficient fatty acid biosynthesis pathway determines efficient surfactin production. The building precursor acetyl-CoA initiates the biosynthesis of fatty acid. The biosynthesis of surfactin is catalyzed through NRPS, initiated by the condensation of fatty acids and Glu. Other constituent amino acids are assembled through the NRPS multi-enzyme complex, comprising adenylation, condensation, and thiolation domains responsible for the activation of amino acids and peptide chain elongation.

## 2.8 Genome evolution of *B. tequilensis*

The enormous genomic data obtained from sequencing of *Bacillus tequilensis* ANSKLAB04 was aligned against the existing top 20 homologous species of *Bacillus* in the NCBI database. The comparison of the number of unique genes and the common genes were analyzed with the top 6 homologous species of *Bacillus* such as *B. subtilis*, *B. vallismortis*, *B. tequilensis* (KCTC 13622), *B. halotolerans*, and *B. mojavensis* [Table 5] [Fig 6].

The phylogenetic tree was constructed from the top 20 homologous *Bacillus* species obtained from a blast search. The orange colors show the gene family expansion and the grey color indicates the gene family contractions between the *Bacillus* species. The corresponding proportions among the total changes are shown in the same colors in the pie chart. Implied divergence dates (in millions of years) are indicated at each node in blue. The most recent common ancestor (MRCA) and the blue color indicate the conserved gene family among the various species of *Bacillus* [Fig 7].

## 2.9 SNP and indel discovery

Our Indel discovery strategy involved mining insertion and deletion polymorphisms from DNA sequencing traces that originally were generated by genome centers for SNP discovery. The obtained mass-sequenced data of *Bacillus tequilensis* ANSKLAB04 were used to search for genetic variation against existing homologous biosurfactant-producing bacteria from GenBank. The present investigation used the existing 5 homologous genomes of bacteria such as *Bacillus tequilensis* KCTC 13622, *Bacillus subtilis*, *Bacillus mojavensis*, *Bacillus vallismortis*, *Bacillus halotolerans*. The number of mapped sites per sample, mapping coverage, the total number of reads, the number of mapped reads, overall mapping ratio, the number of mapped bases, and the average alignment depth were calculated. Table 6 represents the statistics of *Bacillus tequilensis* in comparison with 5 existing homologous bacterial genome which includes *Bacillus tequilensis (KCTC 13622)*, *Bacillus halotolerans*, *Bacillus subtilis*, *Bacillus mojavensis*, *and Bacillus vallismortis*. The number of total reads in all the reference genome was 6,229,938 which were constant in all reference bacteria. The mean depth indicates the number of reads, on average, that were likely to be aligned at a given reference base position in comparison with *Bacillus tequilensis*. However, *Bacillus subtilis was* having 90.39% of mapped

**Table 5. Top 6 organisms homologous to *B. tequilensis* ANSKLAB04.**

| Sl | Organism | Accession | Size in mb | GC% | Genes | Proteins | rRNA | tRNA | Pseudogenes |
|----|----------|-----------|------------|-----|-------|----------|------|------|-------------|
| 01 | *B. tequilensis* ANSKLAB04 | RMVO01000000 | 4.38 | 46.33% | 4724 | 4492 | 28 | 81 | 118 |
| 02 | *B. subtilis* | AL009126.3 | 4.22 | 43.5% | 4536 | 4,237 | 30 | 86 | 88 |
| 03 | *B. vallismortis* | CP026362.1 | 4.28 | 43.80% | 4514 | 4208 | 30 | 87 | 184 |
| 04 | *B. tequilensis* (KCTC 13622) | AYTO00000000.1 | 3.98 | 43.90% | 4167 | 3958 | 7 | 74 | 136 |
| 05 | *B. halotolerans* | CP029364.1 | 4.15 | 43.8% | 4298 | 4032 | 30 | 86 | 145 |
| 06 | *B. mojavensis* | AFSI00000000.1 | 3.96 | 43.7% | 4088 | 3671 | 22 | 81 | 309 |

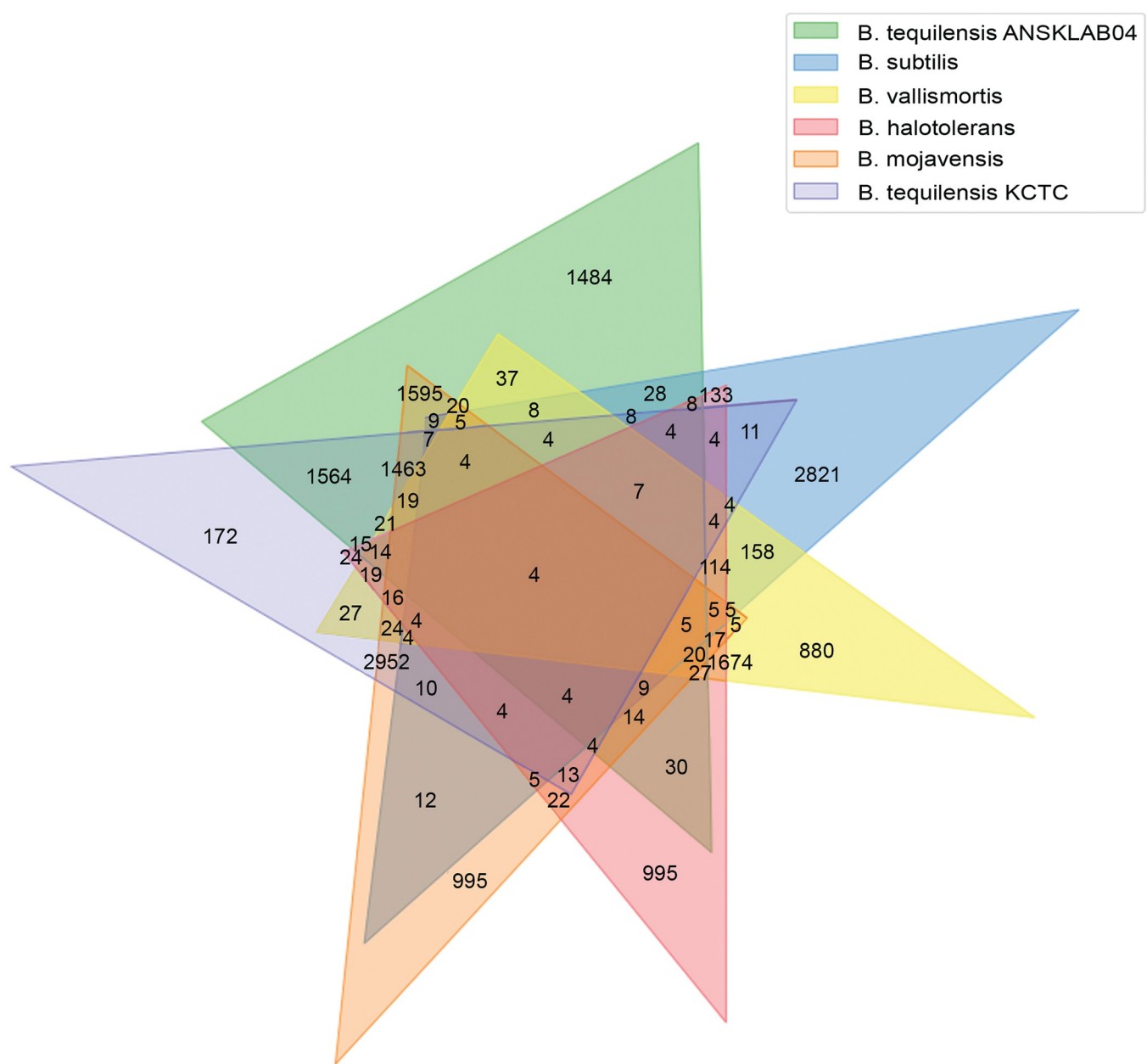

**Fig 6. Comparison of genes of *B. tequilensis* ANSKLAB04 with other species of *Bacillus*.**

read, 786,017,247 mapped bases and 186.45 mean depths which was the highest among the others indicative of better analogy and susceptibility. On the other hand, *Bacillus mojavensis* with reference length 3,957,021, mapped reads 73.19%, mapped bases 555,891,216, and mean depth 140.48 showing the least compatibility with the *Bacillus tequilensis*.

After removing duplicates with Sambamba and identifying variants with SAMTools, information of each variant was gathered and classified by chromosomes or scaffolds. Table 7 shows the summary of the variant calling of *Bacillus tequilensis* ANSKLAB04 against other existing genomes in the database.

Table 7 represents the summary of variant calling of *Bacillus tequilensis* against the existing top 5 homologous references bacterial genome which includes *Bacillus tequilensis (*KCTC

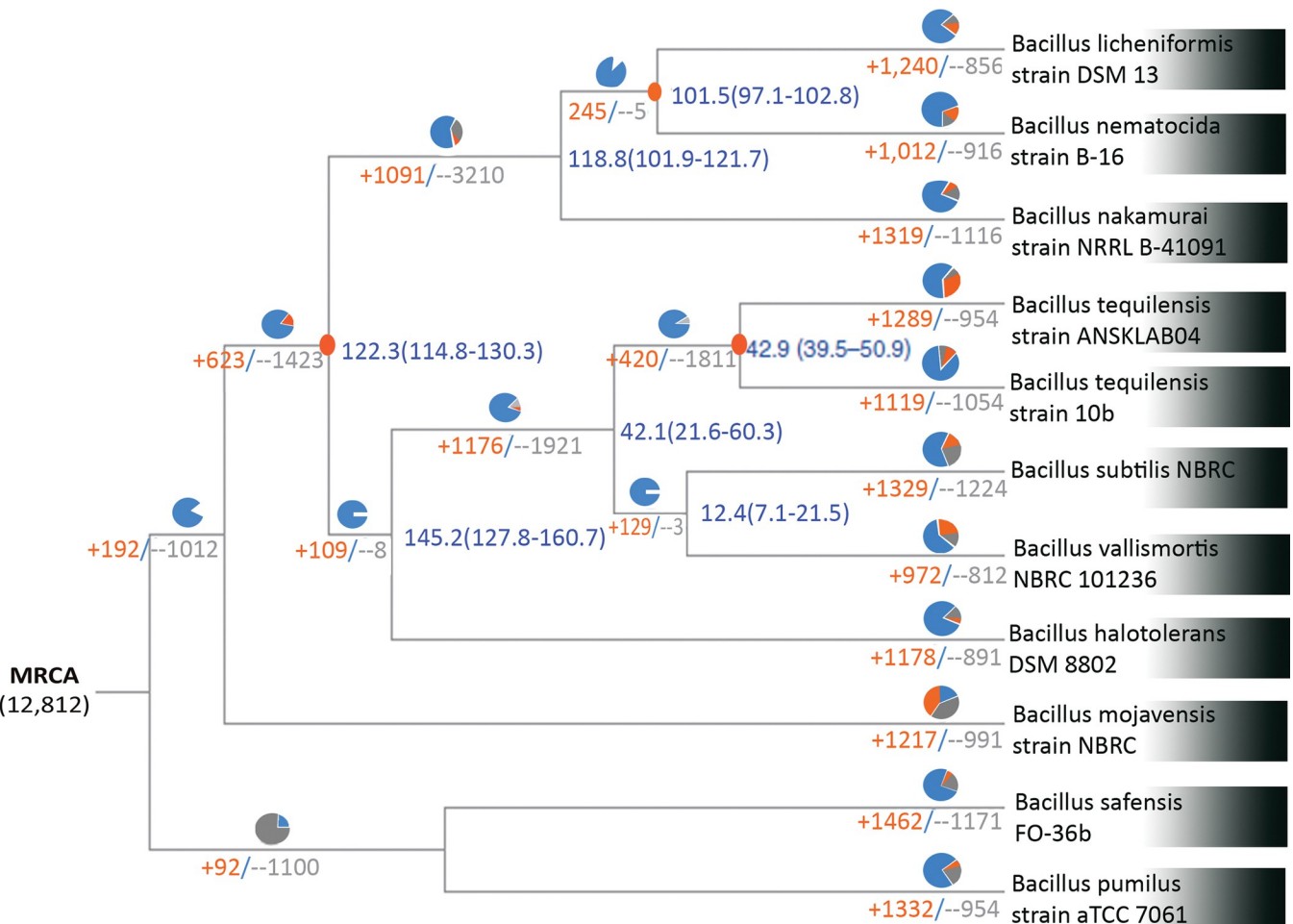

**Fig 7. Phylogenetic affiliation of *Bacillus tequilensis* ANSKLAB04 against other existing species of *Bacillus*.**

13622), *Bacillus halotolerans*, *Bacillus subtilis*, *Bacillus mojavensis* and *Bacillus vallismortis*. Comparison of the whole-genome sequence of *Bacillus tequilensis* in comparison with a reference reveals the number of markers that include single nucleotide polymorphisms (SNPs), inserted and deleted sequences. Fig 8 represents the graphical representation of SNPs and INDEL in which *Bacillus halotolerans* was having the highest number of SNPs i.e. 347,175 [Fig 8D] whereas *Bacillus tequilensis* KCTC 13622 was having the maximum number of insertions and deletions i.e. 841 and 653 respectively [Fig 8A]. Meanwhile, *Bacillus subtilis* were having fewer SNPs, insertions, and deletions [Fig 8B].

**2.9.1 Base change count.** Table 8 and Fig 9 represent the base change count on every SNPs of *Bacillus tequilensis* against the existing 5 homologous reference bacterial genomes

**Table 6. Mapped data statistics of *Bacillus tequilensis* ANSKLAB04 against other homologus existing bacterial reference genome.**

| Ref Genome | Ref Length | Mapped sites (> = 1x) | Total reads | Mapped reads | Mapped bases | Mean Depth |
|---|---|---|---|---|---|---|
| *Bacillus tequilensis* KCTC 13622 | 3,981,302 | 3,510,212 (88.17%) | 6,229,938 | 5,236,833 (84.06%) | 702,871,686 | 176.54 |
| *Bacillus halotolerans* | 4,154,245 | 3,377,421 (81.3%) | 6,229,938 | 4,720,660 (75.77%) | 576,944,088 | 138.88 |
| *Bacillus subtilis* | 4,215,606 | 3,850,277 (91.33%) | 6,229,938 | 5,631,246 (90.39%) | 786,017,247 | 186.45 |
| *Bacillus mojavensis* | 3,957,021 | 3,251,746 (82.18%) | 6,229,938 | 4,559,964 (73.19%) | 555,891,216 | 140.48 |
| *Bacillus vallismortis* | 4,286,362 | 3,466,929 (80.88%) | 6,229,938 | 4,988,614 (80.07%) | 665,216,980 | 155.19 |

**Table 7. Summary of variant calling of *Bacillus tequilensis* ANSKLAB04 against existing species of *Bacillus*.**

| Ref Genome | Library name | Number of SNPs | Number of insertions | Number of deletions |
|---|---|---|---|---|
| *Bacillus tequilensis* KCTC 13622 | SRR8203917(*Bacillus tequilensis* ANSKLAB04) | 261,227 | 841 | 653 |
| *Bacillus subtilis* | SRR8203917(*Bacillus tequilensis* ANSKLAB04) | 47,864 | 496 | 452 |
| *Bacillus vallismortis* | SRR8203917(*Bacillus tequilensis* ANSKLAB04) | 272,438 | 746 | 604 |
| *Bacillus halotolerans* | SRR8203917(*Bacillus tequilensis* ANSKLAB04) | 347,175 | 671 | 625 |
| *Bacillus mojavensis* | SRR8203917(*Bacillus tequilensis* ANSKLAB04) | 338,879 | 692 | 640 |

which includes *Bacillus tequilensis (KCTC 13622) strain* [Fig 9A], *Bacillus subtilis* [Fig 9B], *Bacillus vallismortis* [Fig 9C], *Bacillus halotolerans* [Fig 9D], *Bacillus mojavensis* [Fig 9E].

**2.9.2 Transition and transversion information.** The number of transition (Ts) and transversion (Tv), and the Ts/Tv ratio were calculated using the base change count. Base

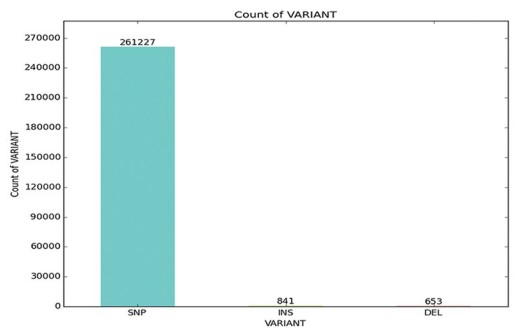

A) Bacillus tequilensis ANSKLAB04  VS  Bacillus tequilensis KCTC 13622

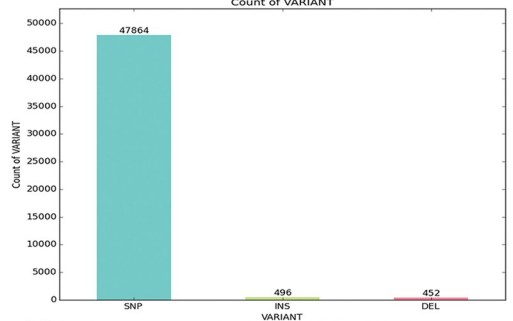
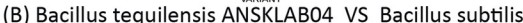

(B) Bacillus tequilensis ANSKLAB04  VS  Bacillus subtilis

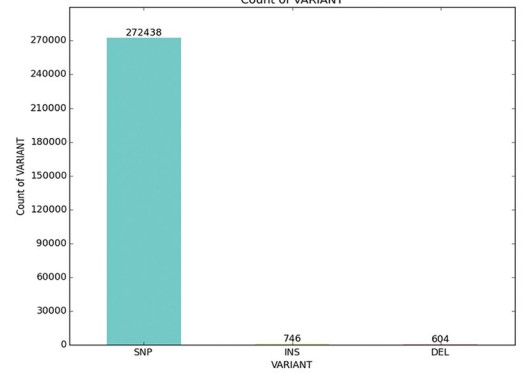
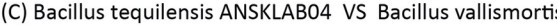

(C) Bacillus tequilensis ANSKLAB04  VS  Bacillus vallismortis

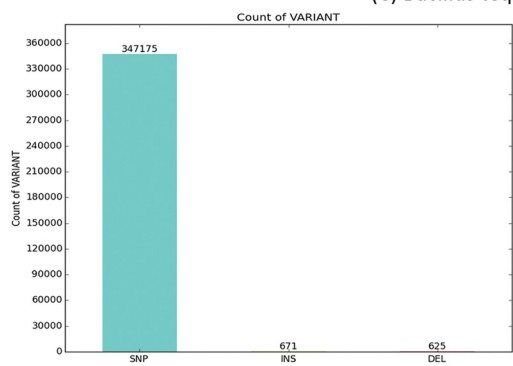

(D) Bacillus tequilensis ANSKLAB04  VS  Bacillus halotolerans

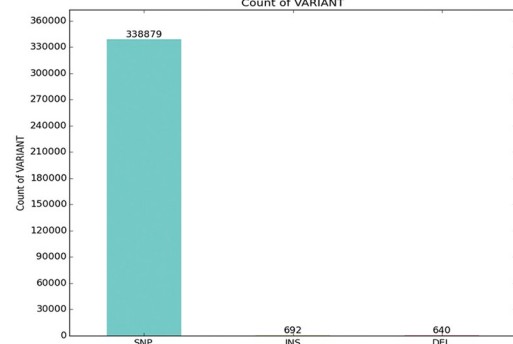

(E) Bacillus tequilensis ANSKLAB04  VS  Bacillus mojavensis

**Fig 8. SNP/Insertion/Deletion count.**

**Table 8. Base count change.**

| | | | | | | | |
|---|---|---|---|---|---|---|---|
| **Bacillus tequilensis ANSKLAB04 vs Bacillus tequilensis KCTC 13622** | | | | | | | |
| Library Name | Ref | | A | | | C | |
| | Alt | T | G | C | A | T | G |
| SRR8203917 | | 13,677 | 39,995 | 10,498 | 13,741 | 44,344 | 8,431 |
| Library Name | Ref | | G | | | T | |
| | Alt | A | T | C | A | G | C |
| | | 44,784 | 13,579 | 8,575 | 13,462 | 10,271 | 39,870 |
| **Bacillus tequilensis ANSKLAB04 vs Bacillus subtilis** | | | | | | | |
| Library Name | Ref | | A | | | C | |
| | Alt | T | G | C | A | T | G |
| SRR8203917 | | 2,391 | 8,082 | 2,021 | 1,997 | 8,227 | 1,142 |
| Library Name | Ref | | G | | | T | |
| | Alt | A | T | C | A | G | C |
| SRR8203917 | | 8,252 | 2,088 | 1,176 | 2,412 | 1,977 | 8,099 |
| **Bacillus tequilensis ANSKLAB04 vs. Bacillus vallismortis** | | | | | | | |
| Library Name | Ref | | A | | | C | |
| | Alt | T | G | C | A | T | G |
| SRR8203917 | | 14,704 | 40,958 | 10,767 | 15,363 | 45,556 | 9,450 |
| Library Name | Ref | | G | | | T | |
| | Alt | A | T | C | A | G | C |
| SRR8203917 | | 45,463 | 14,764 | 9,510 | 14,616 | 10,525 | 40,762 |
| **Bacillus tequilensis ANSKLAB04 vs. Bacillus halotolerans** | | | | | | | |
| Library Name | Ref | | A | | | C | |
| | Alt | T | G | C | A | T | G |
| SRR8203917 | | 18,913 | 51,771 | 17,789 | 17,062 | 53,943 | 14,151 |
| Library Name | Ref | | G | | | T | |
| | Alt | A | T | C | A | G | C |
| SRR8203917 | | 53,584 | 16,973 | 14,202 | 19,145 | 17,655 | 51,987 |
| **Bacillus tequilensis ANSKLAB04 vs. Bacillus mojavensis** | | | | | | | |
| Libray name | Ref | | A | | | C | |
| | Alt | T | G | C | A | T | G |
| SRR8203917 | | 18,917 | 51,594 | 17,012 | 16,395 | 51,610 | 13,798 |
| Library Name | Ref | | G | | | T | |
| | Alt | A | T | C | A | G | C |
| SRR8203917 | | 51,821 | 16,592 | 13,723 | 18,824 | 17,310 | 51,282 |

changes (DNA substitution) are of two types. Interchanges of purines (A <-> G), or pyrimidines (C <-> T) are transitions, while interchanges of a purine for pyrimidine bases, and vice versa, are transversions. Although there are twice as many possible transversions, transitions are more common than transversions due to differences in structural characteristics. Generally, transversions are more likely to cause amino acid sequence changes. [Table 9] represents the transition and transversion information of *Bacillus tequilensis* against 5 existing homologous reference bacterial genome which includes *Bacillus tequilensis (KCTC 13622)*, *Bacillus halotolerans*, *Bacillus subtilis*, *Bacillus mojavensis*, *and Bacillus vallismortis*, and [Fig 10] represents the proportional pie chart of Transversion and transition distribution. The transition/transversion ratio between homologous strands of DNA is generally about 2, but it is typically elevated in coding regions, where transversions are more likely to change the underlying

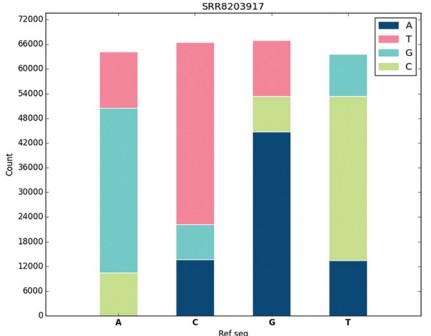

A) Bacillus tequilensis ANSKLAB04  VS  Bacillus tequilensis KCTC 13622

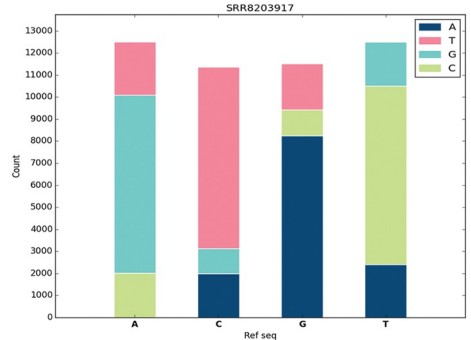

(B) Bacillus tequilensis ANSKLAB04  VS  Bacillus subtilis

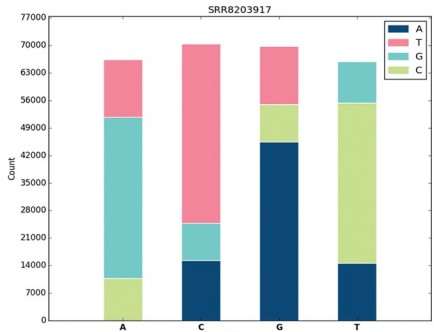

(C) Bacillus tequilensis ANSKLAB04  VS  Bacillus vallismortis

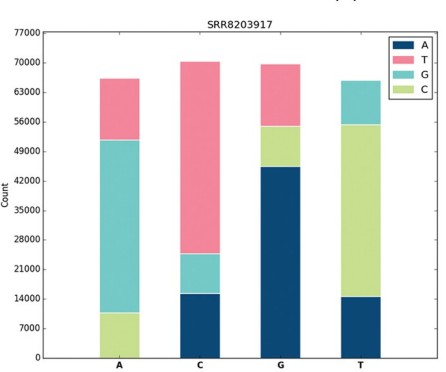

(D) Bacillus tequilensis ANSKLAB04  VS  Bacillus halotolerans

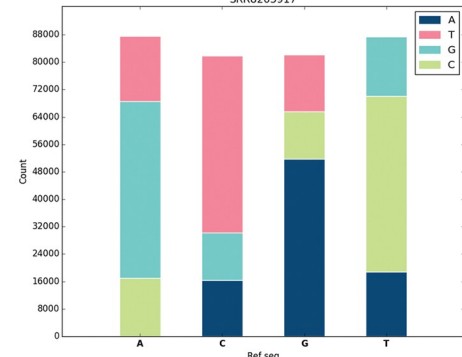

(E) Bacillus tequilensis ANSKLAB04  VS  Bacillus mojavensis

**Fig 9. Base change count of each sample.**

amino acid and thus possibly lead to a fatal mutation in the translated protein. *Bacillus halotolerans* was having a maximum number of total SNPs counts hence their number of transition and transversion counts was also more i.e. 211,285 and 135,890 respectively but the ratio

**Table 9. Transition, transversion information table.**

| Ref Genome | Library Name | Total SNP Count | Transition | Transversion | Ts/Tv |
|---|---|---|---|---|---|
| *Bacillus tequilensis* KCTC 13622 | SRR8203917(*Bacillus tequilensis* ANSKLAB04) | 261,227 | 168,993 | 92,234 | 1.83% |
| *Bacillus subtilis* | SRR8203917(*Bacillus tequilensis* ANSKLAB04) | 47,864 | 32,660 | 15,204 | 2.15% |
| *Bacillus vallismortis* | SRR8203917(*Bacillus tequilensis* ANSKLAB04) | 272,438 | 172,739 | 99,699 | 1.73% |
| *Bacillus halotolerans* | SRR8203917(*Bacillus tequilensis* ANSKLAB04) | 347,175 | 211,285 | 135,890 | 1.55% |
| *Bacillus mojavensis* | SRR8203917(*Bacillus tequilensis* ANSKLAB04) | 338,879 | 206,308 | 132,571 | 1.56% |

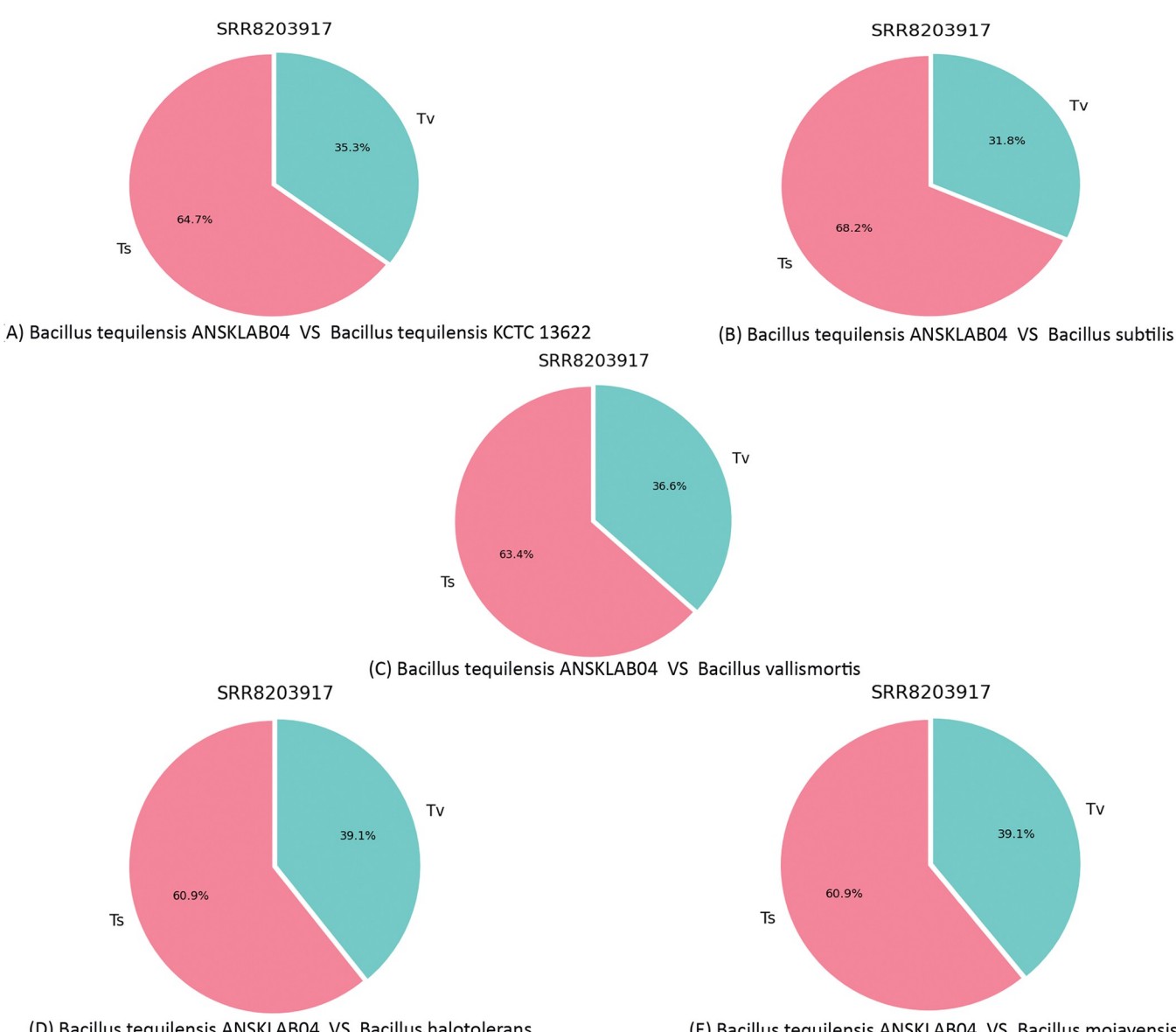

**Fig 10. Transition, transversion proportion.**

percentage of Ts/Tv was 1.55% which was estimated by pairwise sequence comparison. On the other side, *Bacillus subtilis* was having the lowest count of total SNPs, Transition, and Transversion but had the highest Ts/Tv ratio i. e 2.15%. Transition indicative number of A to T and C to G conversion or interchange and vice-versa whereas transversion is indicative of A to C or A to G or T to C or T to G or vice-versa as shown in [Fig 10]. *Bacillus subtilis* was having more transitions in comparison with *Bacillus tequilensis*(Fig 10B) i.e. 68.2%. The number of transversions was more in *Bacillus halotolerans* and *Bacillus mojavensis* i.e. 39.1%. However, in all 5 reference genomes in comparison with *Bacillus tequilensis*, the count percentage of Transition was more than compared to transversion (Fig 10D and 10E]. Transitions are less likely to result in amino acid substitutions and are therefore more likely to persist as "*silent substitutions*" in populations as single nucleotide polymorphisms (SNPs).

**Table 10. Annotation type count of *Bacillus tequilensis* ANSKLAB04 by comparing *Bacillus tequilensis* KCTC 13622.**

| Library name | Type of annotation | Count | Ratio |
|---|---|---|---|
| SRR8203917 | upstream_gene_variant | 256,198 | 98.27% |
| | downstream_gene_variant | 3,561 | 1.37% |
| | intergenic_region | 780 | 0.3% |
| | synonymous_variant | 128 | 0.05% |
| | missense_variant | 40 | 0.02% |
| | splice_region_variant & non_coding_transcript_exon_variant | 5 | 0.0% |
| | splice_region_variant & stop_retained_variant | 5 | 0.0% |
| | disruptive_inframe_insertion | 1 | 0.0% |
| | initiator_codon_variant | 1 | 0.0% |
| | frameshift_variant | 1 | 0.0% |

## 2.10 Variant annotation

To find out the annotation information such as amino acid changes by variants, SnpEff was used. Since genes usually have multiple transcripts, a single variant can have different effects on different transcripts. Tables 8 and 9 show the number of variants per type (based on the representative transcript), and brief explanations about the variant type, respectively. Tables 10–15 shows the top 10 types of variant annotations of *Bacillus tequilensis* ANSKLAB04 by comparing with *Bacillus tequilensis* KCTC, *Bacillus subtilis*, *Bacillus vallismortis*, *Bacillus halotolerans*, *Bacillus mojavensis*.

Table 10 represents the comparison of the annotation type count of *Bacillus tequilensis* ANSKLAB04 when aligned with *Bacillus tequilensis* (KCTC 13622). There were various types

**Table 11. Annotation type count of *Bacillus tequilensis* ANSKLAB04 by comparing *Bacillus subtilis*.**

| Library name | Type of annotation | Count | Ratio |
|---|---|---|---|
| SRR8203917 | upstream_gene_variant | 47,287 | 96.92% |
| | downstream_gene_variant | 1,098 | 2.25% |
| | intergenic_region | 363 | 0.74% |
| | synonymous_variant | 31 | 0.06% |
| | missense_variant | 10 | 0.02% |
| | initiator_codon_variant | 2 | 0.0% |
| | disruptive_inframe_insertion | 1 | 0.0% |

**Table 12. Annotation type count of *Bacillus tequilensis* ANSKLAB04 by comparing *Bacillus vallismortis*.**

| Library name | Type of annotation | Count | Ratio |
|---|---|---|---|
| SRR8203917 | upstream_gene_variant | 270,075 | 98.67% |
| | downstream_gene_variant | 3,200 | 1.17% |
| | intergenic_region | 295 | 0.11% |
| | synonymous_variant | 100 | 0.04% |
| | missense_variant | 46 | 0.02% |
| | splice_region_variant & stop_retained_variant | 6 | 0.0% |
| | splice_region_variant & non_coding_transcript_exon_variant | 2 | 0.0% |
| | disruptive_inframe_insertion | 1 | 0.0% |
| | initiator_codon_variant | 1 | 0.0% |

**Table 13. Annotation type count of *Bacillus tequilensis* ANSKLAB04 by comparing *Bacillus halotolerans*.**

| Library name | Type of annotation | Count | Ratio |
|---|---|---|---|
| SRR8203917 | upstream_gene_variant | 340,311 | 97.74% |
| | downstream_gene_variant | 5,892 | 1.69% |
| | intergenic_region | 1,794 | 0.52% |
| | synonymous_variant | 128 | 0.04% |
| | missense_variant | 59 | 0.02% |
| | splice_region_variant & stop_retained_variant | 10 | 0.0% |
| | initiator_codon_variant | 1 | 0.0% |
| | bidirectional_gene_fusion | 1 | 0.0% |
| | initiator_codon_variant & non_canonical_start_codon | 1 | 0.0% |

of annotation found in *Bacillus tequilensis* ANSKLAB04 when aligned with *Bacillus tequilensis* (KCTC 13622). There upstream gene variant was having a maximum ratio of 98.27% with 256,198 indicative of a sequence variant located at 5' of a gene whereas the downstream gene variant was indicative of a sequence variant located at 3' of a gene which was 3,561(1.37%). There was only 1 count of frameshift variant which indicated a disruption of the translational reading frame because the number of nucleotides inserted or deleted was not a multiple of three which was almost negligible.

Table 11 represents the annotation type count of *Bacillus tequilensis* ANSKLAB04 when aligned with *Bacillus subtilis*. There were 7 types of annotations found in *Bacillus tequilensis* ANSKLAB04 when aligned with *Bacillus subtilis*, which include upstream gene variant, downstream gene variant, intergenic region, synonymous variant, missense variant, initiator codon variant, and disruptive inframe insertion. There upstream gene variant was having a maximum ratio of 96.92% with 47,287 indicative of a sequence variant located at 5' of a gene whereas the downstream gene variant is indicative of a sequence variant located at 3' of a gene which was 1,098 (2.25%). Here synonymous variant count was 31 (0.06%) which was indicative of a sequence variant where there is no resulting change to the encoded amino acid.

Table 12 represents the annotation type count of *Bacillus tequilensis* ANSKLAB04 when aligned with *Bacillus vallismortis*. There were 9 types of annotations found in *Bacillus tequilensis* ANSKLAB04 when aligned with *Bacillus vallismortis*. There upstream gene variant was having a maximum ratio of 98.67% with 270,075 indicative of a sequence variant located at 5' of a gene whereas the downstream gene variant was indicative of a sequence variant located at 3' of a gene which was 3,200 (1.17%).

Table 13 represents the annotation type count of *Bacillus tequilensis* ANSKLAB04 when aligned with *Bacillus halotolerans*. There were various types of annotation found in *Bacillus*

**Table 14. Annotation type count of *Bacillus tequilensis* ANSKLAB04 by comparing *Bacillus mojavensis*.**

| Library name | Type of annotation | Count | Ratio |
|---|---|---|---|
| SRR8203917 | upstream_gene_variant | 331,498 | 97.66% |
| | downstream_gene_variant | 7,427 | 2.19% |
| | intergenic_region | 312 | 0.09% |
| | synonymous_variant | 133 | 0.04% |
| | missense_variant | 37 | 0.01% |
| | splice_region_variant&stop_retained_variant | 11 | 0.0% |
| | splice_region_variant & non_coding_transcript_exon_variant | 6 | 0.0% |
| | initiator_codon_variant | 2 | 0.0% |
| | initiator_codon_variant&non_canonical_start_codon | 1 | 0.0% |

**Table 15. Annotation type information.**

| Type of annotation | Description | Impact |
|---|---|---|
| coding_sequence_variant | The variant hits a CDS. | MODIFIER |
| chromosome | A large part (over 1% or 1,000,000 bases) of the chromosome was deleted. | HIGH |
| duplication | Duplication of a large chromoome segment (over 1% or 1,000,000 bases). | HIGH |
| inversion | Inversion of a large chromoome segment (over 1% or 1,000,000 bases). | HIGH |
| coding_sequence_variant | One or many codons are changed. | LOW |
| inframe_insertion | One or many codons are inserted (e.g.: An insert multiple of three in a codon boundary). | MODERATE |
| disruptive_inframe_insertion | One codon is changed and one or many codons are inserted (e.g.: An insert of size multiple of three, not at codon boundary). | MODERATE |
| inframe_deletion | One or many codons are deleted (e.g.: A deletion multiple of three at codon boundary). | MODERATE |
| disruptive_inframe_insertion | One codon is changed and one or more codons are deleted (e.g.: A deletion of size multiple of three, not at codon boundary). | MODERATE |
| downstream_gene_variant | Downstream of a gene (default length: 5K bases). | MODIFIER |
| exon_variant | The variant hits an exon (from a non-coding transcript) or a retained intron. | MODIFIER |
| exon_loss_variant | A deletion removes the whole exon. | HIGH |
| exon_loss_variant | Deletion affecting part of an exon. | HIGH |
| duplication | Duplication of an exon. | HIGH |
| duplication | Duplication affecting part of an exon. | HIGH |
| inversion | Inversion of an exon. | HIGH |
| inversion | Duplication affecting part of an exon. | HIGH |
| frameshift_variant | Insertion or deletion causes a frame shift (e.g.: An indel size is not multple of 3). | HIGH |
| gene_variant | The variant hits a gene. | MODIFIER |
| feature_ablation | Deletion of a gene. | HIGH |
| duplication | Duplication of a gene. | MODERATE |
| gene_fusion | Fusion of two genes. | HIGH |
| gene_fusion | Fusion of one gene and an intergenic region. | HIGH |
| bidirectional_gene_fusion | Fusion of two genes in opposite directions. | HIGH |
| rearranged_at_DNA_level | Rearrengment affecting one or more genes. | HIGH |
| intergenic_region | The variant is in an intergenic region. | MODIFIER |
| Conserved_intergenic_variant | The variant is in a highly conserved intergenic region. | MODIFIER |
| intragenic_variant | The variant hits a gene, but no transcripts within the gene. | MODIFIER |
| intron_variant | Variant hits and intron. Technically, hits no exon in the transcript. | MODIFIER |
| conserved_intron_variant | The variant is in a highly conserved intronic region. | MODIFIER |
| miRNA | Variant affects an miRNA. | MODIFIER |
| missense_variant | Variant causes a codon that produces a different amino acid (e.g.: Tgg/Cgg, W/R). | MODERATE |
| initiator_codon_variant | Variant causes start codon to be mutated into another start codon (the new codon produces a different AA). (e.g.: Atg/Ctg, M/L (ATG and CTG can be START codons)) | LOW |
| stop_retained_variant | Variant causes stop codon to be mutated into another stop codon (the new codon produces a different AA). (e.g.: Atg/Ctg, M/L (ATG and CTG can be START codons)) | LOW |
| protein_protein_contact | Protein-Protein interacion loci. | HIGH |
| structural_interaction_variant | Within protein interacion loci (e.g. two AA that are in contact within the same protein, prossibly helping structural conformation). | HIGH |
| rare_amino_acid_variant | The variant hits a rare amino acid thus is likely to produce protein loss of function. | HIGH |
| splice_acceptor_variant | The variant hits a splice acceptor site (defined as two bases before exon start, except for the first exon). | HIGH |
| splice_donor_variant | The variant hits a Splice donor site (defined as two bases after coding exon end, except for the last exon). | HIGH |
| splice_region_variant | A sequence variant in which a change has occurred within the region of the splice site, either within 1–3 bases of the exon or 3–8 bases of the intron. | LOW |
| splice_region_variant | A variant affective putative (Lariat) branch point, located in the intron. | LOW |
| splice_region_variant | A variant affective putative (Lariat) branch point from U12 splicing machinery, located in the intron. | MODERATE |
| stop_lost | Variant causes stop codon to be mutated into a non-stop codon (e.g.: Tga/Cga, */R). | HIGH |

*(Continued)*

**Table 15.** (Continued)

| Type of annotation | Description | Impact |
|---|---|---|
| 5_prime_UTR_premature start_codon_gain_variant | A variant in 5'UTR region produces a three base sequence that can be a START codon. | LOW |
| start_lost | Variant causes start codon to be mutated into a non-start codon (e.g.: aTg/aGg, M/R). | HIGH |
| stop_gained | Variant causes a STOP codon (e.g.: Cag/Tag, Q/*). | HIGH |
| synonymous_variant | Variant causes a codon that produces the same amino acid (e.g.: Ttg/Ctg, L/L). | LOW |
| start_retained | Variant causes start codon to be mutated into another start codon (e.g.: Ttg/Ctg, L/L (TTG and CTG can be START codons)). | LOW |
| stop_retained_variant | Variant causes stop codon to be mutated into another stop codon (e.g.: taA/taG, */*). | LOW |
| transcript_variant | The variant hits a transcript. | MODIFIER |
| feature_ablation | Deletion of a transcript. | HIGH |
| regulatory_region_variant | regulatory_region_variant The variant hits a known regulatory feature (non-coding). | MODIFIER |
| upstream_gene_variant | Upstream of a gene (default length: 5K bases). | MODIFIER |
| 3_prime_UTR_variant | Variant hits 3'UTR region. | MODIFIER |
| 3_prime_UTR_truncation + exon_loss | The variant deletes an exon which is in the 3'UTR of the transcript. | MODERATE |
| 5_prime_UTR_variant | Variant hits 5'UTR region. | MODIFIER |
| 5_prime_UTR_truncation + exon_loss_variant | The variant deletes an exon which is in the 5'UTR of the transcript. | MODERATE |

Description for the Table 15

Type of annotation: Sequence ontology which allows to standardize the terminology used for assessing sequence changes and impact.

Description: Detailed description of the effect (annotation).

Impact: Effects are categorized by 'impact': {High, Moderate, Low, Modifier}. These are pre-defined categories to help users find more significant variants.

HIGH: The variant is assumed to have a high (disruptive) impact on the protein, probably causing protein truncation, loss of function or triggering nonsense-mediated decay.

MODERATE: A non-disruptive variant that might change protein effectiveness.

LOW: Assumed to be mostly harmless or unlikely to change protein behavior.

MODIFIER: Usually non-coding variants or variants affecting non-coding genes, where predictions are difficult or there is no evidence of impact.

*tequilensis* ANSKLAB04 when aligned with *Bacillus halotolerans*. There upstream gene variant was having a maximum ratio of 97.74% with 340,311 indicative of a sequence variant located at 5' of a gene whereas the downstream gene variant was indicative of a sequence variant located at 3' of a gene which was 5,892 (1.69%).

Table 14 represents the annotation type count of *Bacillus tequilensis ANSKLAB04* when aligned with *Bacillus mojavensis*. There were various types of annotation found in *Bacillus tequilensis* ANSKLAB04 when aligned with *Bacillus mojavensis*. There upstream gene variant was having a maximum ratio of 97.66% with 331,498 indicative of a sequence variant located at 5' of a gene whereas the downstream gene variant was indicative of a sequence variant located at 3' of a gene which was 7,427 (2.19%).

Variant calling tool SnpEff reports the putative variant impact to make it easier and faster to categorize and prioritize variants. However, impact categories must be used with care as they were created only to help and simplify the filtering process. There is no way to predict whether a HIGH impact or a LOW impact variant is the one producing a phenotype of

**Table 16. Library concentration estimation using Qubit.**

| Sample ID | Qubit Conc. (ng/μl) | Vol (l) | Yield(ng) | Nextflex Barcode | Barcode Sequence | qPCR conc. (nM) |
|---|---|---|---|---|---|---|
| SO_4915_Bt1_ ePCR1_IL_WGS | 3.92 | 12 | 47.04 | 4 | GCCAAT | 7.092 |

interest. The results of the variant calling of *Bacillus tequilensis* ANSKLAB04 when aligned with *Bacillus tequilensis* KCTC 13622, *Bacillus subtilis*, *Bacillus vallismortis*, *Bacillus halotolerans*, *Bacillus mojavensis* are provided in [S3–S7 **Tables**] and annotation type information are provided in [Table 15] [**S8 Table**].

## 3. Discussion

In the present investigation, we have introduced a high-quality draft genome sequence of *Bacillus tequilensis*, the first genome sequence of biosurfactant producing *Bacillus tequilensis* has been determined. Biosurfactant-producing microbes have potential applications in various biotechnology, biodegradation and pharmaceutical industries. The whole genome sequence of biosurfactant-producing *Bacillus tequilensis* will provide a foremost resource to start exploring the genes and gene products involved in biosurfactant synthesis. The genome sequence of *Bacillus tequilensis* obtained in the present investigation will be a key resource for the development of new concepts and techniques in genetic engineering such as molecular marker-assisted breeding and large-scale production of biosurfactant microbes for bioremediation.

### 3.1 Data availability

FASTQC: http://www.bioinformatics.babraham.ac.uk/projects/fastqc

   Trimmomatic: http://www.usadellab.org/cms/?page=trimmomatic

   SPAdes Assembler: https://github.com/ablab/spades

   SSPACE: http://www.baseclear.com/bioinformatics-tools/

   NCBI Prokaryotic Genome Annotation Pipeline:
   https://www.ncbi.nlm.nih.gov/genome/annotation_prok/

   COG: https://www.ncbi.nlm.nih.gov/COG/

   MISA: https://webblast.ipk-gatersleben.de/misa/

   BWA: http://bio-bwa.sourceforge.net/

   Sambamba: http://lomereiter.github.io/sambamba/

   SAM tools: http://samtools.sourceforge.net/

   SnpEff: https://pcingola.github.io/SnpEff/

## 4. Methods

### 4.1 Sample collection and DNA isolation

The strain was isolated from Chilika Lake, a brackish water lagoon, spread over the Puri, Khurda, and Ganjam districts of Odisha state on the east coast of India [12]. Water samples were collected from oil-contaminated sites of Chilika Lake, Odisha, India (latitude and longitude: 19.8450 N 85.4788 E), the largest brackish water lagoon in India. Various organisms were isolated and purified on culture plates and were then enriched in the mineral salt medium (MSM). MSM gives the nutrient condition for the production of biosurfactants by the organisms which were then screened for their biosurfactant production by various screening tests and the emulsification index was calculated. Identification of organisms was performed based on biochemical, macroscopic, and microscopic characteristics. The organism with the best emulsification index was then subjected to optimization for the production of biosurfactants for the factors affecting the production. Optimization was studied with the emulsification index calculated with each affecting factor. In a previous study, this organism was then subjected to 16S rRNA sequencing for the identification of the genus and species [12]. The DNA was isolated by Phenol/Chloroform (PCl) genomic DNA extraction method [12, 19]. The bacterial cell pellet obtained after centrifugation was subjected to DNA isolation. The DNA

concentration and purity were checked with a nanodrop spectrophotometer and qubit fluorometer [20].

## 4.2 Materials used in the study

Whole Genome Sequencing kits such as NEXTFlex DNA Sequencing Kit (Cat # 5140–02), NEXTFlex DNA Barcodes– 48 (Cat # 514104), HighPrep™ PCR (Magbio, #AC-60050), High Sensitivity Bioanalyzer Chips (Agilent, #5067–4626), Nuclease free water (Ambion, #AM9939), Covaris™ S220 System (Life Technologies, #4465653), Covaris™ microTUBE AFA (Life Technologies, #520045), Low Melting Agarose (Invitrogen, #16520100), MinElute Gel Extraction kit (QIAGEN, #28604), 50X TAE Buffer (MP Biomedical, Cat #TAE50X01), Qubit® dsDNA HS Assay Kit (Invitrogen, Cat # Q32854) were used in the present investigation.

## 4.3 Library preparation and genome sequencing

Library preparation was performed using the NEXTFlex DNA library protocol outlined in the "NEXTFlex" DNA sample preparation guide (Cat # 5140–02). In brief, genomic DNA was sheared to generate fragments of approximately 300-500bpin a Covaris micro Tube with the E220 system (Covaris, Inc., Woburn, MA, USA). The fragment size distribution was checked using Agilent Bioanalyzer (Agilent Technologies, Santa Clara, CA) with High Sensitivity DNA Kit (Agilent Technologies) according to the manufacturer's instructions. The resulting fragmented DNA was cleaned up using HighPrep beads (MagBio Genomics, Inc, Gaithersburg, Maryland). These fragments were subjected to end-repair, A-tailing, and ligation of the Illumina multiplexing adaptors using the NEXTFlex DNA Sequencing kit as per the manufacturer's instruction [21].

The resulting ligated DNA was cleaned up using HighPrep beads (MagBio Genomics, Inc, Gaithersburg, Maryland)and size selected (400–600bp) on 2% low melting agarose gel and cleaned using MinElute column (QIAGEN, India). These adapter-ligated fragments were subjected to 10 rounds of PCR (denaturation at 98˚C for 2 min, cycling (98˚C for the 30S, 65˚C for 30S and 72˚C for 1 min) and a final extension at 72˚C for 5 min) using primers provided in the NEXTFlex DNA Sequencing kit(Perkin Elmer). The PCR products were purified using HighPrep beads. Quantification and size distribution of the prepared library was determined using Qubit flourometer (Table 16) and the Agilent High Sensitivity DNA Kit (Agilent Technologies) respectively according to the manufacturer's instructions [Fig 11]. Illumina Paired-end sequencing was performed using NextSeq 500: 150*2. The following adapters were used for sequencing (Illumina, Inc) [21].

Adapter details: Universal Adapter 5′
`AATGATACGGCGACCACCGAGATCTACACTCTTTCCCTACACGACGCTCTTCCGATCT 3′`

Adapter, Index 5′ `GATCGGAAGAGCACACGTCTGAACTCCAGTCAC[INDEX]`
`ATCTCGTATGCCGTCTTCTGCTTG 3′`

## 4.4 Whole genome de-novo assembly and analysis

The obtained sequence raw reads were checked for quality control using the FASTQC tool [22]. The quality of the raw reads was checked through the various modules provided by the FASTQC tool. Among the modules, per base sequence quality and tile sequence quality modules were studied to validate the quality of the data for further analysis. The low-quality reads were excluded from the analysis using Trimmomatic (v0.36) [23]. The filtered De-novo assembly of Illumina paired-end data was assembled using SPAdes—v3.13.0 genome assembler—an open-source algorithm for De-novo assembly [24]. SPAdes assembler is intended for de-novo assembly after error correction of sequenced reads. Assembled contigs were further scaffolded using the SSPACE program [25]. A genome map was constructed using Circos [26].

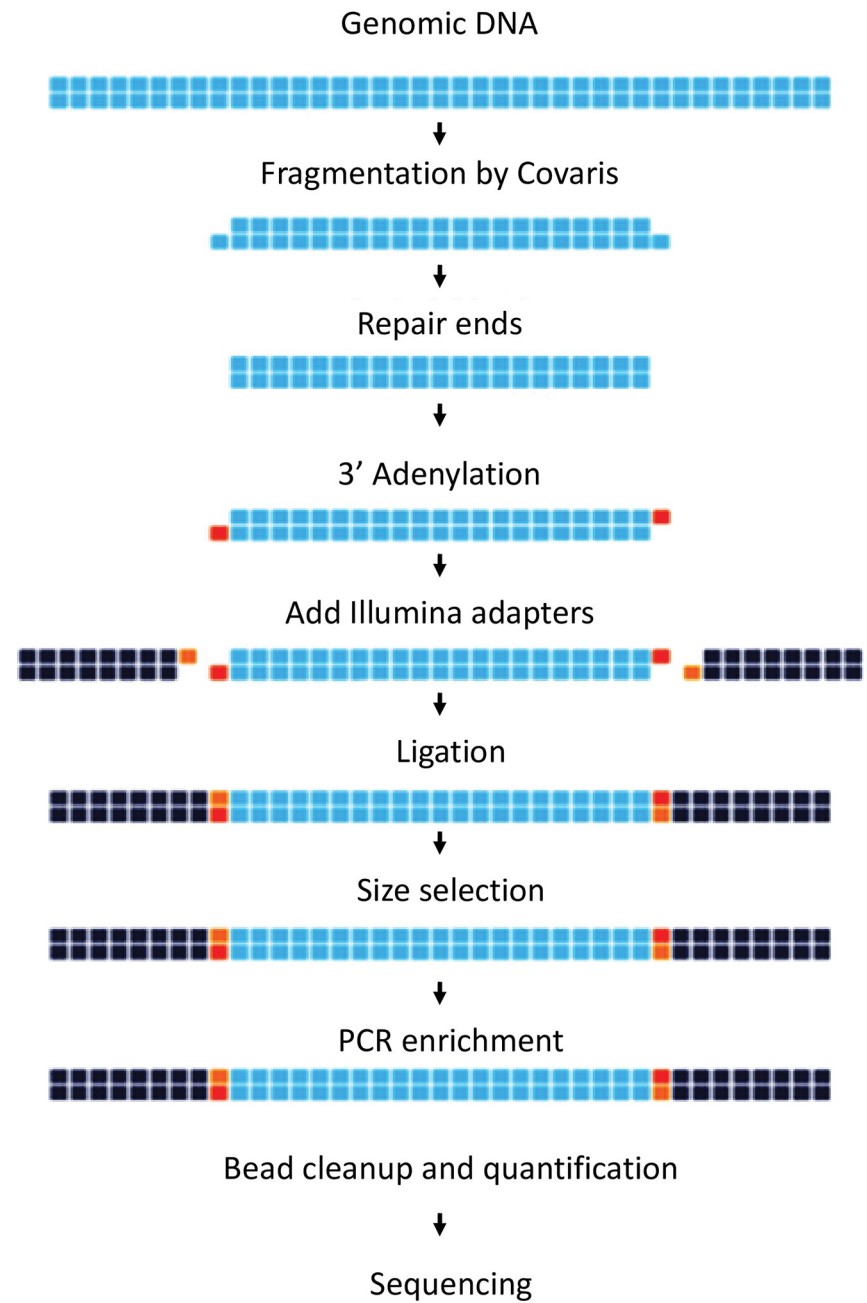

**Fig 11. Work flow for whole genome library preparation using NEXTFlex DNA sample preparation guide.**

## 4.5 Whole genome annotation and GO analysis

NCBI Prokaryotic Genome Annotation Pipeline (PGAP) version 4.8 was used to annotate the whole genome sequence [27]. Pathway Analysis was done by using the KAAS Server. *Bacillus subtilis subsp 168* was taken as a reference organism for pathway analysis using KAAS server [28]. The functions of the predicted ORFs were categorized by comparison with the COG database [29]. Venn diagram was constructed using matplotlib—venn in Python [30]. Simple Sequence Repeats (SSR) were identified in each transcript sequence using the MISA Perl script [31].

## 4.6 Variant calling and variant annotation

Variant Calling of *Bacillus tequilensis* was performed by aligning with the top *5* existing homologous reference bacterial genome which includes *Bacillus tequilensis*(KCTC 13622), *Bacillus halotolerans*, *Bacillus subtilis*, *Bacillus mojavensis, and Bacillus vallismortis*. The present investigation used BWA (Burrows-Wheeler Aligner)-MEM for the alignment of *Bacillus tequilensis* against the top 5 homologous genomes [32]. During mapping, duplicated reads can falsely cause erroneous data to stand out. To prevent such errors, the Sambamba tool was used to remove the duplicate reads [32]. Duplicate reads are identified using mapping information such as start position, and CIGAR string [33]. SAMTools was used to manipulate the SAM/ BAM files that come out as a result of mapping [34]. In resequencing analysis, it is especially used for finding out variant information by calculating genotype likelihood from every position within the sample of analysis. Variant annotation was performed using SnpEff (v4.3t) [35]. SnpEff annotates the possible effects (on genes) that can be caused by variants identified through mapping. The present study used SnpEff to generate the Genes and transcripts affected by the variant, the location of the variants, and the information on how the variant affects the protein synthesis (e.g. generating a stop codon).

## Supporting information

**S1 Table. Genome annotation data.**
(XLSX)

**S2 Table. Gene annotation: Subcategory, subsystem and role.**
(XLSX)

**S3 Table. Variant calling of *Bacillus tequilensis* ANSKLAB04 against *Bacillus tequilensis* (KCTC 13622).**
(XLSX)

**S4 Table. Variant calling of *Bacillus tequilensis* ANSKLAB04 against *Bacillus halotolerans*.**
(XLSX)

**S5 Table. Variant calling of *Bacillus tequilensis* ANSKLAB04 against *Bacillus subtilis*.**
(XLSX)

**S6 Table. Variant calling of *Bacillus tequilensis* ANSKLAB04 against *Bacillus mojavensis*.**
(XLSX)

**S7 Table. Variant calling of *Bacillus tequilensis* ANSKLAB04 against *Bacillus vallismortis*.**
(XLSX)

**S8 Table. Variant Annotation information.**
(XLSX)

## Acknowledgments

The authors are thankful to Eminent Biosciences and LeGene Biosciences Pvt Ltd, Indore, India for 16S rRNA sequencing and Whole Genome Sequencing, and De novo assembly of the bacterium.

Availability of data and materials

This whole-genome shotgun project has been deposited in GenBank/ENA/DDBJ under the Accession: RMVO00000000. The short-read sequences have been deposited under BioProject

Accession: PRJNA498807, BioSample Accession: SAMN10335300, and SRA Accession: SRX5023292.

WGS URL: https://www.ncbi.nlm.nih.gov/nuccore/RMVO00000000

Bioproject URL: https://www.ncbi.nlm.nih.gov/bioproject/?term=PRJNA498807

Biosample URL: https://www.ncbi.nlm.nih.gov/biosample/SAMN10335300

SRA URL: https://www.ncbi.nlm.nih.gov/sra/?term=SRX5023292

Genes/Proteins: https://www.ncbi.nlm.nih.gov/Traces/wgs/RMVO01?display=contigs

## Author Contributions

**Conceptualization:** Anuraj Nayarisseri, Sanjeev Kumar Singh.

**Data curation:** Anuraj Nayarisseri, Sanjeev Kumar Singh.

**Formal analysis:** Anuraj Nayarisseri.

**Investigation:** Anuraj Nayarisseri, Sanjeev Kumar Singh.

**Methodology:** Anuraj Nayarisseri, Sanjeev Kumar Singh.

**Project administration:** Anuraj Nayarisseri, Sanjeev Kumar Singh.

**Software:** Anuraj Nayarisseri.

**Supervision:** Anuraj Nayarisseri, Sanjeev Kumar Singh.

**Validation:** Anuraj Nayarisseri, Sanjeev Kumar Singh.

**Visualization:** Anuraj Nayarisseri, Sanjeev Kumar Singh.

**Writing – original draft:** Anuraj Nayarisseri, Sanjeev Kumar Singh.

**Writing – review & editing:** Anuraj Nayarisseri, Sanjeev Kumar Singh.

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
