## [Editor Report · Decision Letter 0]

5 Jan 2023

PONE-D-22-19370Whole-Genome Sequencing and Annotation of Biosurfactant Producing Bacillus tequilensisPLOS ONE

Dear Dr. Singh,

Thank you for submitting your manuscript to PLOS ONE. After careful consideration, we feel that it has merit but does not fully meet PLOS ONE’s publication criteria as it currently stands. Therefore, we invite you to submit a revised version of the manuscript that addresses the points raised during the review process.

We look forward to receiving your revised manuscript.

Kind regards,

Pankaj Kumar Arora

Academic Editor

PLOS ONE

Journal Requirements:

Additional Editor Comments (if provided):

1. Title should be changed as "Genome analysis of biosurfactant producing bacterium, Bacillus tequilensis.

2. All microorganisms name should be italics.

3. Data related to genome-based taxonomy including average nucleotide identity and digital DNA-DNA hybridization with closely related type strains should be provided.

4. These lines are incorrect "we concluded that Bacillus tequilensis produces biosurfactants and the

novel isolate was deposited in GenBank with accession number KU529483". You have submitted nucleotide sequence of 16S rRNA gene to the Genbank not bacteria. Similar mistakes are in whole manuscript.

5. Manuscript is too lengthy; it should be concise. It should be focused only on biosurfactant related genes.

6. Experimental data related to biosurfactant production in this strain are necessary.
---

## [Author Response · Author response to Decision Letter 0]

24 Feb 2023

RESPONSE LETTER

Ref.: Whole Genome Sequencing and Annotation of Biosurfactant Producing Bacillus tequilensis

Dear Editors,

 We appreciated the review of our manuscript and the suggestions given to make it suitable to PLOS One journal which is in high quality standards. As requested, this letter responds to each point raised by the reviewers. Please, kindly note that changes were marked up in the manuscript and are transcribed herein:

Editor Comments

1. Title should be changed as "Genome analysis of biosurfactant producing bacterium, Bacillus tequilensis.

Response: We appreciate the careful evaluation of our manuscript and the recommendations made to raise the quality of it. As advised by the editor, we would like to let the editor know that the title of the manuscript has been changed to "Genome study of biosurfactant generating bacteria, Bacillus tequilensis." Thanks.

2. All microorganisms name should be italics.

Response: We value the in-depth analysis of our manuscript and the recommendations made for enhancing its quality. We would want to let the editor know that all of the microbe names have been changed to italics as per the editor's suggestion. Thanks.

3. Data related to genome-based taxonomy including average nucleotide identity and digital DNA-DNA hybridization with closely related type strains should be provided.

Response: We value the thorough analysis of our manuscript and the suggestions offered to improve its quality. We would like to inform the editor that in section 2.1 first paragraph, we have added the genome-based taxonomy including ANI (Average Nucleotide Identity) analysis and digital DNA-DNA hybridization/ Genome – Genome comparison by GGDC(Genome Genome Distance Calculator) with closely related type strains of Bacillus tequilensis as suggested by the editor. Thanks. 

4. These lines are incorrect "we concluded that Bacillus tequilensis produces biosurfactants and the novel isolate was deposited in GenBank with accession number KU529483". You have submitted nucleotide sequence of 16S rRNA gene to the Genbank not bacteria. Similar mistakes are in whole manuscript.

Response: We value the thorough analysis of our manuscript and the suggestions offered to improve its quality. We would want to inform the editor that we have changed this throughout the manuscript as suggested by the editor. I appreciate you pointing out this error.

5. Manuscript is too lengthy; it should be concise. It should be focused only on biosurfactant related genes.

Response : We appreciate the in-depth evaluation of our manuscript and the recommendations made to raise its quality. Since the work involves a whole genome analysis and complete annotation of a bacteria, we would want to let the editor know that we made every effort to keep the paper as concise as possible. As a result, the text became a little lengthy. We made an effort to discard a few information, such SSR identification. Let us know if there are any other analysis need to be deleted. Looking forward to hear for the suggesion.

6. Experimental data related to biosurfactant production in this strain are necessary.

Response : We value the thorough analysis of our manuscript and the suggestions offered to improve its quality. We would like to inform the editor that, we have carried out a number of experimental studies including the Haemolysis test, oil spreading test, CTAB agar plate test, Drop collapse test etc from which we have concluded that the Bacillus tequilensis isolate used in this investigation produces biosurfactants. This study has already been published, so we've used it as a source in result session 2.2 in the present manuscript.The present study is a continuation of the earlier investigation.The publication can be access from the folowign link.

https://link.springer.com/article/10.1007/s13762-018-2089-9

---

## [Decision Letter · Decision Letter 1]

7 May 2023

Genome analysis of biosurfactant producing bacterium, Bacillus tequilensis

PONE-D-22-19370R1

Dear Dr. Singh,

We’re pleased to inform you that your manuscript has been judged scientifically suitable for publication and will be formally accepted for publication once it meets all outstanding technical requirements.

Kind regards,

Pankaj Kumar Arora

Academic Editor

PLOS ONE

Additional Editor Comments (optional):

Reviewers' comments:

Reviewer's Responses to Questions

**Comments to the Author**

1. If the authors have adequately addressed your comments raised in a previous round of review and you feel that this manuscript is now acceptable for publication, you may indicate that here to bypass the “Comments to the Author” section, enter your conflict of interest statement in the “Confidential to Editor” section, and submit your "Accept" recommendation.

Reviewer #1: All comments have been addressed

2. Is the manuscript technically sound, and do the data support the conclusions?

Reviewer #1: Yes

3. Has the statistical analysis been performed appropriately and rigorously? 

Reviewer #1: Yes

4. Have the authors made all data underlying the findings in their manuscript fully available?

Reviewer #1: Yes

5. Is the manuscript presented in an intelligible fashion and written in standard English?

Reviewer #1: Yes

6. Review Comments to the Author

Reviewer #1: All comments and queries were addressed appropriately. The manuscript can be accepted in the present form.

7. PLOS authors have the option to publish the peer review history of their article (what does this mean?). If published, this will include your full peer review and any attached files.

Reviewer #1: No

---

## [Editor Report · Acceptance letter]

22 May 2023

PONE-D-22-19370R1 

Genome analysis of biosurfactant producing bacterium, *Bacillus tequilensis*. 

Dear Dr. Singh:

I'm pleased to inform you that your manuscript has been deemed suitable for publication in PLOS ONE. Congratulations! Your manuscript is now with our production department. 

Kind regards, 

on behalf of

Dr. Pankaj Kumar Arora 

Academic Editor

PLOS ONE